# Physically-based distributed hydrological model calibration based on a short period of streamflow data: case studies in four Chinese basins

Wenchao Sun[1,2], Yuanyuan Wang[1,2], Guoqiang Wang[1,2] Xingqi Cui [1,2], Jingshan Yu[1], Depeng Zuo[1,2]
Zongxue Xu [1,2]

[1] College of Water Sciences, Beijing Normal University, Xinjiekouwai Street 19, Beijing 100875, China
[2] Joint Center for Global Change Studies (JCGCS), Beijing 100875, China

*Correspondence to*: Jingshan Yu (jingshan@bnu.edu.cn)

**Abstract.** Physically-based distributed hydrological models are widely used for hydrological simulations in various
environments. As with conceptual models, they are limited in data-sparse basin by the lack of streamflow data for calibration. Short periods of observational data (less than 1 year) may be obtained from fragmentary historical records of past-existed gauging stations or from temporary gauging during field surveys, which might be of values for model calibration. However, unlike lumped conceptual model, such an approach hasn't been explored sufficiently for physically-based distributed models. This study explored how the use of limited continuous daily streamflow data might support the application of a physically-
based distributed model in data-sparse basins. The influence of the length of observation period on the calibration of the widely applied Soil and Water Assessment Tool model was evaluated in four Chinese basins with differing climatic and geophysical characteristics. The evaluations were conducted by comparing calibrations based on short periods of data with calibrations based on data from a 3-year period, which were treated as benchmark calibrations of the four basins, respectively. To ensure the differences in the model simulations solely come from differences in the calibration data, the
Generalized Likelihood Uncertainty Analysis scheme was employed for the automatic calibration and uncertainty analysis. In the four basins, contrary to the common understanding of the need for observations over a period of several years, data records with lengths of less than 1 year were shown to calibrate the model effectively, i.e. performances similar to the benchmark calibrations were achieved. The models of wet Jinjiang Basin and Donghe Basin could be effectively calibrated using a shorter data record (1 month), compared with the dry Heihe Basin and upstream Yalongjiang Basin (6 months). Even
though the four basins are very different, when using 1-year or 6-month (covering a whole dry season or rainy season) data, the results show that data from wet seasons and wet years are generally more reliable than data from dry seasons and dry years, especially for the two dry basins. The results demonstrated that this idea could be a promising approach to the problem of calibration of physically-based distributed hydrological models in data-sparse basins and findings from the discussion in this study are valuable for assessing the effectiveness of short period data for model calibration in real world application.
**Key words:** Physically-Based Distributed Hydrological Model; Length of observation data; Model calibration: Data-sparse basin

## 1 Introduction

Globally, flood and droughts are the two most prevalent natural disasters, considered to have affected 140 million people annually, on average, between 2005 and 2014 (United Nations Offices for Disaster Risk Reduction, 2016). Mitigating the possible damages associated with these disasters relies on precise forecasting in term of timing and scale (Callahan et al, 1999; McEnery et al, 2005). Hydrological models are tools commonly used for simulating the water cycle at basin scale and for predicting streamflow at the basin outlet, which represents the integrated output of all the hydrological processes within a basin. Many parameters of hydrological models are conceptual without explicit physical meaning, which makes it necessary to identify parameter values through model calibration based on streamflow data (Gupta, 2005). For physically-based model, although values of parameters with explicit physical meaning can be measured, the scale of measurement and model simulation is different, which makes it difficult to apply measured values to hydrological models directly. Also, these measurements require intensive field survey, which are not available in most researches. Therefore, usually parameters of such model are also obtained from model calibration based on streamflow data. However, because of resource constraints (e.g., financial and human resources), there has been a general decline in the networks designed to monitor streamflow (Wohl et al., 2012), especially in developing countries, which has become a major obstacle to the applications of hydrological models in basins where streamflow data are sparse (Hrachowitz et al, 2013).

The usual approach regarding data-sparse basins is regionalization, which estimates model parameters using information from similar gauged basins. One major concern with regionalization is prediction uncertainty, which is determined by the degree of similarity and by the method chosen to describe the similarity (Sivapalan, 2003). To reduce the uncertainty introduced by regionalization, many researchers have tried to improve parameter estimation by introducing limited information from ungauged basins. For example, Viviroli and Seibert (2015) combined short-term streamflow observations with parameter regionalization and showed that parameter identifications could be improved compared with using information only from donor basins. Many recent works have tried to use available in situ or remote sensing observations of hydrological processes other than streamflow for model calibration, e.g., soil moisture (e.g., Silvestro et al., 2015; Vrugt et al., 2002), evapotranspiration (Vervoort et al., 2014; Winsemius et al., 2008), groundwater level(e.g., Khu et al., 2008), as a new direction to solve the calibration problem. These studies have shown promising performances for identifying parameters that describe the processes being measured. However, none of these observations has the similar capability as streamflow data for constraining hydrological model parameters. Another appealing approach is the use of river water surface area, width, or stage derived from remote sensing as a surrogate of streamflow for model calibration (e.g., Revilla-Romero et al., 2015; Sun et al., 2015; Getirana, 2010); however, such an approach depends on the availability of effective satellite observations. Furthermore, the reported higher simulation uncertainty in comparison with calibration based on streamflow data is another concern (Sun et al., 2010, 2012).

From the above, it is clear that streamflow observations play a critical role in identifying hydrological model parameters. For an ungauged basin, although a long time series of observations is unavailable, short-period records of streamflow or

occasional observations from field surveys might be obtainable. If such data are to be used for calibration, to know how many observations are needed to calibrate model parameter is important. It is usually suggested that streamflow records covering several years are necessary (Yapo et al., 1996); however, several researchers have attempted to challenge this common understanding using discontinuous or a short-period records of less than 1 year in basins within different climatic regions (e.g., Perrin et al., 2007; Kim and Kaluarachchi, 2009; Seibert and Beven, 2009; Tada and Beven, 2012). For conceptual models, these researches indicated that with observations of the order of several scores, reasonable parameter estimates could be derived. And model performance similar to those obtained from calibrations using records covering several years could be obtained, highlighting the possibility that calibration with limited numbers of observations is a promising alternative to the classical regionalization approach. For hydrological simulations or predictions in changing environments, when the model is expected to evaluate influences of change in climate or the basin's physical characteristics to the water cycle, physically-based distributed hydrological models are usually preferred, because of their better description of the spatial heterogeneity and details of the water cycle at the basin scale (Finger et al, 2012; Wu and Liu, 2012). However, the use of limited observations to address the calibration problem of such models in ungauged basins has been discussed rarely in the literature, probably because of the complexity of model structures and the corresponding considerable demands for computation time.

The objective of this study is to explore how short-period of daily continuous streamflow observations might support the calibration of a physically-based distributed model in data-sparse basins. In real world, such observations might be obtained from fragmentary historical records of past-existed gauging stations or from short-period field surveys. The commonly used Soil and Water Assessment Tool (SWAT) model was adopted for the investigation. Previous research has shown that the requirements of calibration data differ significantly among basins (e.g., Liden and Harlin, 2000). Therefore, we selected four basins with different climatic conditions (two in humid regions and the other two in dry regions) to improve the generality of our findings. The evaluation relies on comparison with the conventional calibration using observations covering several years, which was adopted as the benchmark calibration. The evaluation requires an objective calibration and uncertainty analysis framework to ensure the differences among the calibration results derived solely from the differences in the observations. Considering this issue, the Generalized Likelihood Uncertainty Estimation (GLUE) (Beven and Binley, 1992; Freer and Beven, 1996) method was used for the model calibration, for which all the settings during the calibration were verifiable and satisfying the requirements of the evaluation. By reducing the number of observations used in the model calibration in a designed manner and by comparing each with the benchmark calibration, the influence of the length of observational records on the calibration could be analyzed and the feasibility of using limited data discussed.

## 2 Materials and method

### 2.1 Hydrological model

SWAT is a popular physically-based distributed hydrological model developed by the United States Department of Agriculture. It operates on a daily-time step, and it is capable of simulating the water cycle and transportation of sediment and pollutants at the basin scale. The model is fully integrated with geographic information system (GIS). Based on a river network derived from a digital elevation model, the study basin can be discretized into many subbasins. Moreover, based on GIS data of the soil type and land cover, each subbasin can be separated into several unique hydrological response units for describing the heterogeneity in runoff generation. The hydrological processes considered in the model include precipitation, interception, infiltration, evapotranspiration, snowmelt, surface runoff, percolation, baseflow, and flow movement in river channels. Because of the complex model structure, many parameters need to be identified via calibration. Further details about the SWAT model are available in Arnold et al.(1998) and Gassman et al.(2007).

### 2.2 Calibration and uncertainty analysis method

Considering the objective of this study, manual calibration is not feasible for the comparison of calibrations because it relies on subjective judgments about model performance (Madsen, 2003). Therefore, an automatic calibration procedure that optimizes an objective function by searching parameter spaces to find combinations reflecting the characteristics of target basin was required (Muleta and Nicklow, 2005). Another concern is the phenomenon of equifinality (Beven, 2001) that many very different parameter sets might exhibit similar performances. Thus, it is necessary to quantify the uncertainty introduced by equifinality for the evaluation. Here, the GLUE method was employed as the automatic calibration and uncertainty analysis scheme. It was integrated to the SWAT model in the calibration package SWAT-CUP (SWAT Calibration Uncertainty Procedures) (Yang et al., 2008). To describe the equifinality in a quantitative manner, it regards all those parameter sets performing better than a predefined threshold as behavioral parameter sets, for which the corresponding simulations with weights assigned based on performance are then used to produce an ensemble simulation. Several subjective options must be made when using the GLUE method, but they are made explicitly and they can be examined at any time (Beven and Binley, 1992). For different calibrations, if all subjective settings except the calibration data remain the same, the GLUE method can ensure that differences in the calibration results derive purely from the different observations used in the calibration, which is ideal for the comparison needed for the evaluation of this study. Here, the procedure for the implementation of the GLUE method was follows:

1. Generate random parameter sets. Usually, the prior information about parameter distributions is unknown, and therefore assuming uniform distributions is reasonable (Beven and Freer, 2001). Then, the Latin hypercube sampling scheme is adopted to generate parameter sets randomly from parameter space.

2. Select behavioral parameter sets. A likelihood measure is defined to quantify the degree of goodness with which each parameter set can reproduce the observations. Then, based on a threshold set by the modeler, the good parameter sets (named behavioral parameter sets) are selected. Here, the Nash-Sutcliffe efficiency (NSE) was used as the likelihood measure:

$$NSE = 1 - \frac{\sum(Q_{obs,i} - Q_{sim,i})}{\sum(Q_{obs,i} - Q_{obs,avg})} \tag{1}$$

where $Q_{obs,i}$ (m³/s) and $Q_{sim,i}$ (m³/s) represent the observed and simulated streamflow, respectively, at time step $i$, and $Q_{obs,avg}$ (m³/s) is the average value of the streamflow observations.

3. Calculate the behavioral parameter sets' posterior likelihood. Every identified behavioral set is included to make an ensemble simulation. The posterior likelihood of each set, i.e., the weight of the streamflow simulation of each behavioral parameter set in the ensemble simulation, is computed based on the Bayes equation:

$$L_p[\theta | Q_{obs}] = CL[\theta | Q_{obs}]L_o[\theta] \tag{2}$$

where $L_o[\theta]$ is the prior likelihood of parameter set $\theta$, under the assumption of a uniform prior distribution (which is the same value for all sets), $L[\theta|Q_{obs}]$ is the NSE that quantifies the performance of reproducing $Q_{obs}$, and $C$ is a scaling factor makes unity the sum of posterior likelihood for all behavioral parameter sets.

4. Make an ensemble prediction. At each time step $t$, the cumulative distribution of the simulation is calculated:

$$P_t(Q_t < q) = \sum_{i=1}^{m} L_p[\theta_i | Q_{t,i} < q] \tag{3}$$

where $P (Q_t<q)$ is the cumulative probability of the simulated streamflow $Q_t$ less than an arbitrary value $q$, $L_p[\theta_i]$ is the posterior likelihood of set $\theta_i$, for which the simulated streamflow is less than $q$, and $m$ is the amount of the parameter sets that satisfy the condition of $Q_{t,i} < q$. The streamflow corresponding to the lower 2.5% and upper 97.5% quantiles of the posterior distribution at each time step consists of the lower and upper limits of the ensemble simulation, respectively. The predicted streamflow corresponding to the best performing parameter set (judged from likelihood) is treated as the best estimate of streamflow.

## 2.3 Study basins

We used 4 basins located in China (Figure 1) to test the method. The basins are spread over the country to ensure that various hydrological, climatic and geophysical conditions are included in our study. They located in different climatic regions and characteristics of topography, annual precipitation and temperature are quite different.

The Jinjiang Basin is located on the west coast of the Taiwan Straits in Fujian Province, China. The area of the basin is 5629 km². The river system has two major tributaries that flow from mountainous area of the north, join at Shuangxikou, and then flow to the low plain region in the southeast (elevation ranges from 50 to 1366 m). The dominant land covers are forest and crop land, and the main soil types are paddy soil, red soil, and yellow soil. The basin is in a subtropical marine

monsoon climatic region, with warm dry winters and hot rainy summers. Annual precipitation ranges from 1000 to 1800 mm, most of which falls in summer. The hydrological modelling was conducted for the upstream area of the Shilong gauging station.

The Donghe River is one of the major tributaries of the Pengxi River in the upstream region of the Three Gorges Reservoir. The length of the mainstream is about 106 km, and the drainage area is 1,089 $km^2$. The main land covers are cropland, shrub and pasture, and the main soil types are flat stone yellow sandy soil and lime yellow clay. The basin is in a warm wet subtropical monsoon climate region. Annual precipitation ranges from 1100 to 1500 mm. Most of precipitation falls in summer. The hydrological model was calibrated and validated by the streamflow data in Wenquan gauging station.

The Heihe Basin is in the arid northwest of China. It is the second largest inland basin in China with an area about 128,900 $km^2$. From the southern mountainous region to the northern high-plain area, the elevation decreases from about 5000 to 1000 m. The hydrological simulation was executed for the upstream mountainous region of Yingluoya gauging station, encompassing an area of around 8,843 $km^2$. The elevation of the study area varies from around 4700m in the headwater region to around 1700 m at Yingluoya station. The primary land cover types are forest, grassland, and Gobi, and alpine meadow soil and frost desert soil occupy more than 74% of the basin area. The region has an inland continental climate with cold dry winters and hot summers, with average annual precipitation around 400mm.

The Yalongjiang River is originated in the Tibetan plateau, which is the largest tributary to the Jinshajiang River in the upper Yangtze River. The hydrological modelling was conducted in the upstream region of Ganzi streamflow station, for which the elevation ranges from 3400 to 6021. The area of hydrological simulation by SWAT is 32,535 $km^2$. Plateau meadow is the main soil type, and the shrub meadow is the main land cover type. This basin has a continental plateau climate. Average annual precipitation in recent 50 years is about 520mm, 73% percentage of which concentrated in June to September. A long, cold winter and a cool, wet summer exists in this basin, with strong radiation all over a year.

In the four basins, most precipitation happens in summer. Based on annual precipitation, the four basins were divided into two groups. The Jinjiang and Donghe Basin are considered as representatives of wet basins. And Heihe and Yalongjiang Basin represent dry basins. For each basin, the influence of the observational record length on the calibration will be explored. Then, discussion about the differences among the four basins will be performed to obtain more general insights. The characteristics of the four basins are shown in Table 1. The diversity among the basins is very helpful for making relatively general conclusions from the findings of this study.

## 2.4 Experiment design

Considering the availability of streamflow records, the benchmark calibration for the Jinjiang Basin was made based on full daily observations for 2005–2007 and it was validated using data for 2008-2009. For the other three basins, the benchmark calibrations were also conducted using 3-year continuous daily streamflow observations and the models were validated using 2- or 3-year streamflow data, based on data availability. The details about the calibration and validation period for the benchmark calibrations conducted in the four basins are shown in Table 2. As an initial trial for showing the potential of the

method for distributed models, we sought to explore whether there are records of certain short length or certain number of continuous daily observations could achieve similar performance as benchmark calibration, not to determine whether all records of that specific length can calibrate the model effectively. Simulations by distributed models are time-consuming and the calibration using the GLUE method requires models to be run a large number of times. Therefore, it was hard to follow the studies of conceptual models (e.g. Perrin et al 2007; Seibert and Beven, 2009) that could conduct calibrations many times. Considering the above mentioned two issues, to perform the calibration in manageable times, the experiment of calibration using short period records, which are subsets of the calibration data of the benchmark calibration, was conducted in two stages. In the first stage, three calibrations using 1-year data record that covered both the rainy and dry seasons, and five calibrations using 6-month data record that covered either a rainy season or a dry season were undertaken. The short periods for which corresponding data were used for the calibrations in the first stage are listed in Table 3. If there are calibrations using 6-month data could achieve performances similar to the benchmark calibration, stage two of the experiment was initialized, in which the subsets of 6-month data records were used for calibration to explore the performance of calibration period shorter than six month. Kim and Kaluarachchi (2009) and Yapo et al. (1996) showed that data from high-flow periods are more informative than data from low-flow periods for model calibration. As our study explored the possibility of highest performance of certain lengths of records for calibration, the 3-month, 1-month data and 1-week datasets with highest average streamflow in the 6-month records were employed as the representatives to calibrate the model and conduct the evaluation at these three temporal scales.

Perrin et al. (2007) showed that model performance in the calibration period could be very good when using very limited numbers of observations, because it is easy to fill only a small number of points in the hydrograph. Conversely, the performance in the validation period could be very poor, because there are no observations to constrain the model simulation. Therefore, the evaluation of limited numbers of streamflow data needs to consider the performance in both calibration and validation periods, mostly in the validation period. For each basin, in order to compare model performance of calibration using short period data with the benchmark calibration in an objective manner, the validation periods of these calibrations were made same with the benchmark calibrations. The evaluation of each calibration was performed in terms of the aspects of general performance and simulation uncertainty. The general performance was represented by the *NSE* of the best behavioral parameters set (i.e., the one with the highest likelihood value identified by the calibration data) for the calibration and validation periods. Two indexes were utilized to assess the simulation uncertainty: The *P_factor* is the percentage of observations embraced by the 95% prediction intervals. The *R_factor* is a measure of the average width of 95% simulation intervals

$$R\_factor = \frac{\sum_{i=1}^{m}(Q_{97.5\%,i} - Q_{2.5\%,i})}{m \times \sigma_{Q_{obs}}}$$  (4)

where $Q_{97.5\%,i}$ and $Q_{2.5\%,i}$ represent the 97.5% and 2.5 % quantiles of the simulated streamflow at time step $i$, respectively, $m$ is the total time step of the simulation, and $\sigma_{Qobs}$ is the standard deviation of the streamflow observations. A low value of *R_factor* combined with a high value of the *P_factor* indicates low simulation uncertainty. For the evaluation, we put more weight on *NSE* and *P_factor*, as they are important and explicit for judging the model performance. After *NSE* and *P_factor*, then *R_factor* will be considered and less weight are put.

## 3 Results and discussion

### 3.1 Performances of benchmark calibrations

Before we can apply the model for evaluating the method proposed in this study, the model robustness in the four basins must be examined, through assessing model performance corresponding to the benchmark calibrations. Ten commonly calibrated SWAT parameters from the literature were selected for the automatic calibration using GLUE, and their prior ranges were set based on the recommendation from SWAT-CUP. The parameters and their prior ranges (Table 4) were the same for all calibrations to exclude the influence of parameter uncertainty and ease the calibration comparisons. For each calibration, 10,000 parameter sets were generated randomly using the Latin hypercubic sampling method to run the GLUE scheme. For the Heihe Basin, the threshold for likelihood was set to 0.5. For the Jinjiang Basin, too many parameter sets could result reasonable simulations, and therefore the threshold for likelihood was set to 0.7. The threshold for Donghe Basin and Yalongjiang Basin was 0.5 and 0.4, respectively. For the benchmark calibrations of the four basins, the results of the calibration are summarized in Table 5, and the best simulations and uncertainty bands of the ensemble simulation are shown in Figs. 2 to 5. The NSEs of the best simulation in four cases were satisfactory and they could reproduce the observed hydrographs well. Furthermore, the uncertainty bands covered most of the observations. All these facts indicate that the model applications in the four basins were successful. The results of these calibrations were treated as the benchmarks for each basin. The only difference between the benchmark calibrations and the other calibrations was the calibration data, which were therefore the only cause of the differences in the calibration results.

### 3.2 Evaluation of the Jinjiang Basin case

The performances of the ensemble simulations corresponding to the 1-year and 6-month calibration datasets are shown in Figure 6. For the 1-year period, all three calibrations performed similarly to the benchmark calibration. Figure 7(a) presents the cumulative distribution of the available annual streamflow for Shilong station from 1958 to 2009. For the three year that streamflow data used for benchmark calibration, 2006 is a very wet year and 2005 and 2007 are normal to wet years. The two year of 2008 and 2009 are all dry years. For the benchmark calibration, the model performance in the validation period decreases, compared with calibration period. The decrease in model performance is consistent with other studies (e.g. Todorovic and Plavsic, 2016), in which model efficiency also decrease if calibration period is wetter than validation period. When using only 1-year data for calibration, the performances in validation period are similar to benchmark calibration. On

the 6-month time scale, the corresponding five calibrations exhibited considerable differences: No parameter sets were identified as behavioral parameter set when using calibration data for the period October 2006 to March 2007, indicating that no parameter sets could capture the characteristics of the hydrological processes of that period. The other four records of 6-month could achieve similar performance as benchmark calibration. The second stage of the experiment was undertaken

5    using 3-month (June–August), 1-month (July), and 1-week (July 14–20) datasets with the highest streamflow during April to September 2006. Figure 6 shows that when calibrating the SWAT model using the 1-week dataset, the uncertainty increased and the NSE decreased distinctly in the validation period compared with the benchmark calibration. The calibration using the 1-month dataset still achieved similar performance to benchmark calibration, judging from the indexes. Figure 8 shows that simulated streamflow of best performed parameter sets corresponding to the benchmark calibration, calibration using the 1-

10   month data and the 1-week data. The difference of simulations between the benchmark calibration and calibration using 1-month data is minor. But the difference between the benchmark calibration and calibration using 1-week data is obvious. The latter seems to fail to reproduce streamflow in low flow period, indicating the information content in the observations is not sufficient for model calibration. In summary, it is indicated that in the Jinjiang Basin, it is possible that 1-month's continuous daily observations can contain much of the information content in the 3-year continuous streamflow data for model

calibration.

### 3.3 Evaluation of the Donghe Basin Case

Figure 9 describes the general performance and simulation uncertainty of calibration using 1-year and 6-month data in the Donghe Basin. As long time series of annual streamflow is unavailable, we cannot judge the frequency of annual streamflow in the five years being simulated. Streamflow at the basin outlet is generated by precipitation within the basin. There is a

close relationship between them. Based on this understanding, we use the annual precipitation frequency derived from a national climatic dataset with spatial resolution of 5 km, which was developed by the Land-Atmosphere Interaction Research Group at Beijing Normal University (available at: http://globalchange.bnu.edu.cn/research/forcing), as a surrogate of streamflow frequency to infer whether each year is a wet year or a dry year. For the three calibration years, as shown in Figure 7(b), 2002 is a dry year. 2003 and 2004 are wet years. The two validation year, 2005 and 2006 is a wet year and

extremely dry year, respectively. No matter using the 1-year data from a dry year (2002 or 2004) or a wet year (2003), the streamflow in the validation period are all reproduced well. Also all of calibrations using 6-month data, either from rainy seasons or dry seasons, achieve similar performances as the benchmark calibration. Like the case of Jinjiang basin, if some parameter sets are identified as behavioral ones using short period data of 1-year and 6-month, their performances in validation period can resemble the benchmark calibration. The stage two of the evaluation was carried out using the data of

July to September 2003, September 2003, and July 1 to 7, 2003 as the representatives of 3-month, 1-month and 1-week data. There is no parameter set could reach the threshold of likelihood when using the 1-week record. For the other two calibrations, they can all works well as the benchmark calibration. Like the Jinjiang Basin, 1-month data could also calibrate the model successfully in this basin.

### 3.4 Evaluation of the Heihe Basin case

The results of the calibration are shown in Figure 10. The calibrations using 1-year datasets of 2003 and 2005 achieved almost the same performance as the benchmark calibration. For the calibration using data from 2004, the number of identified behavioral parameter sets decreased significantly and the NSE of the best simulation in validation period decreased, indicating the 2004 dataset was less informative than the other 2 years. The cumulative distribution of annual streamflow at Yingluoya Station for 1960–2008 (Fig. 7(c)) indicates that 2004 was an extremely dry year. The other calibration years (2003, 2005) and validation years (2006 to 2008) are all wet years. The limited number of identified behavioral parameter sets derived from calibration using data of 2004 might only fit the situation of this extremely dry year and they might not perform well in other periods. For the calibrations with 6-month dataset, only the wet season of 2003, which was the wettest among the 3 years, demonstrated performance comparable with the benchmark calibration. The performances of the other four calibrations were inferior to that of calibration based on the 3-year dataset. Even the calibration using the dataset of wet season of 2004 fails to identify behavioral parameter sets. In the arid Heihe basin, most rainfall occurs in the summer season. About 75% of total annual streamflow come from the wet period from April to September. Compared with normal year, either in wet season or in dry season of in extremely dry year 2004, the average streamflow decreased. Considering the big contribution to annual total streamflow, the degree of streamflow decrease in the wet period has high possibility to be bigger than the dry season. The runoff generation mechanism in this wet season with extremely low streamflow is very different from normal situation, which made the model cannot capture the essence of variation in streamflow, therefore none of the randomly generated 10,000 parameter sets can reproduce the hydrograph of this wet season with acceptable accuracy. Subsets of data for the wet season of 2003 were selected for the second stage of the experiment. The 3-month, 1-month, and 1-week periods with the highest streamflow were June–August, August, and August 8–14, respectively. None of calibrations based on these datasets achieved similar levels of performance as the benchmark calibration. Based on our evaluation, it is shown that a 6-month dataset could act as a surrogate for 3-year observational period for model calibration in this arid basin.

### 3.5 Evaluation of the Yalongjiang Basin case

The cumulative distribution of annual streamflow at Ganzi station (Figure 7(d)) indicate that, for the calibration period, 2005 is an extremely wet year, 2006 and 2007 are extremely dry years; for the validation period, 2009 is a wet year, 2008 and 2010 are dry years. Figure 11 indicates that, when using 1-year data for calibration, only the wet year 2005 could reach similar level of performance as benchmark calibration. The decreases in model performance when using data of dry year 2006 and 2007 are significant. At the temporal scale of 6-month, the diversity among datasets is high. The 6-month data of rainy season and dry season in the wettest year 2005 could resemble performance of the benchmark calibration. Only one and six parameter sets are identified as behavioral sets when using rainy season data of extremely dry year of 2006 and 2007, respectively. Similar to the Heihe Basin case, it may be caused by the fact that the runoff generation mechanism in these

period differing from normal situation, which made the model fail to capture the substantial processes of streamflow variation. For the observations of October 2006 to March 2007, although some behavioral parameters are gained and model performance at calibration period is satisfying, the calibrated model cannot reproduce the streamflow at validation period with acceptable accuracy. When using wettest 3-month, 1-month and 1-week data for calibration, no behavioral parameter set was identified, indicting these three short period datasets cannot calibrated the model effectively.

## 3.6 Implications for future applications

The results of this study prove that datasets of continuous daily observations covering periods less than 1 year have the potential to calibrate the SWAT model effectively. In the two wet basins, a 1-month dataset of daily streamflow data could achieve calibration results as good as the benchmark calibration. In the two dry basins, calibration using a 6-month dataset could resemble the performance of calibration using the 3-year dataset. This is in accordance with previous research using lumped conceptual models (Tada and Beven, 2012; Perrin et al. 2007; Seibert and Beven, 2009). Even though the distributed model used in this study is more complex, the results still agree with the findings of Liu and Han (2010), i.e., the information content of the calibration data is more important than the length of the dataset, indicating only a dataset covering several months might contains sufficient information for parameter identification. This study clearly demonstrates the value of fragmentary historical records of past-existed gauging stations or temporary gauging during field surveys for calibrating physically-based distributed hydrological model in data sparse basin, at least for basins with a climate characterized by rainy or relatively rainy summer and dry winter and correspondingly streamflow exhibits an annual cycle of high flow and low flow. The paper could inspire more researchers to think about using such dataset to calibrate distributed hydrological models in basins lacking of streamflow data and test it in more well gauged basins to develop more general understanding about when the measurements are most informative for parameter calibration. In the past, this approach didn't draw much attention for solving the calibration problem of distributed models.

When applying the method to real world, the biggest challenge is to judge whether the calibrated model can reflect hydrological characteristics of the simulated data-sparse basin. Many calibrations conducted in this study show that if the model could work well in calibration period, their performance in validation period is also good. Therefore, the phenomenon that some parameter sets are identified behavioral ones based on the comparison between simulation and observations could be considered as one evidence for making the judgement that the short period data is effective for model calibration. However, such judgement should be made with care. When the number of observations becomes lower, our results show that the possibility of good performance in calibration period accompanied by good simulation in validation period decreases. In most calibrations of the two wet basins, such judgement based on model performance in calibration period is valid. However, when the number of observations is too low (e.g., the calibration in Jinjiang Basin using 1-week observations), it may not valid. In the two dry basins, there are several calibrations showing that good performance in calibration period does not ensure good performance in validation period: when using 1-year data for calibration, performance of dry year data is inferior to wet year data. In the Yalongjiang Basin case, the calibrations using 1-year data of dry year 2006, 2007 even fail to

reproduce streamflow in validation period. In the two dry basins, when using 6-month data for calibration, the diversity of model performance is higher than using 1-year data. This might indicate that drier basins require a greater quantity of data for model calibration, which has been proved by the study using a conceptual model (Lidén and Harlin, 2000), because climatic variability is higher and the runoff generation mechanism is more complex than that in wet basins. Generally in the

two dry basins, if the model performance in calibration period is good, 6-month data from wet years or wet periods makes more reliable simulations in validation period than the ones from dry years or dry periods. Kim and Kaluarachchi (2009) demonstrated that data from high-flow periods have greater control on model calibration because they are more informative with regard to parameter identification. In this context, our suggestion is in line with those made by Yapo et al. (1996) and Melsen et al. (2014): Data from wetter periods should be preferred for model calibration.

These findings indicates that, to know the "wetness level" of the short period data, i.e., the records were observed in a wet year or dry year and in a rainy season or dry season, may be helpful to judge whether good simulation could be derived from calibration using a certain short period of observations. In such context, information about annual streamflow frequency and intra-annual streamflow regime at the basin outlet is valuable, as the wetness level of the short period observations can be determined from this information. The coming question is how to get such information in basins lacking streamflow data.

Streamflow at a basin outlet is generated by the precipitation data within the basin. A close relationship between streamflow and precipitation exist in a basin. Precipitation data can be obtained more easily than streamflow data, either from in situ gauging or satellite observations. There are publicly available precipitation products (e.g., Global Historical Climatology Network, data available at: https://www.ncdc.noaa.gov/oa/climate/ghcn-daily; Asian Precipitation-Highly Resolved Observational Data Integration Towards the Evaluation of Water Resources, data available at:

http://www.chikyu.ac.jp/precip/english) with wide temporal coverage and fine spatial resolution that are sufficient to analyse annual precipitation frequency and intra-annual precipitation regime at basin scale, like the 5-km spatial resolution data used in the case of Donghe Basin. Information about the precipitation frequency can work as surrogates of annual streamflow frequency and intra-annual streamflow regime to determine the wetness level of certain short period streamflow data in real applications and correspondingly the performance of calibrated model could be indirectly assessed. To develop a general

understanding of whether the information content in certain limited calibration data record is sufficient to obtain robust parameter values, further researches similar to this one, using distributed model in large number of well-gauged basins with differing characteristics, is required. Such study need to generate many samples of short period observations from available streamflow data and then the samples are used to calibrate the model. A reasonable sampling strategy is needed for this kind of research. Our study shows that, to some extent, the wetness level of short period data is related to the performance of

calibrated model. Therefore, considering the wetness level of data in the sampling strategies may be valuable to obtain general guideline on when the short period observations are informative for model calibration.

## 4 Conclusions

This study was an initial evaluation of the possibility of calibrating physically-based distributed hydrological models using limited streamflow data, which could be extracted from available fragmentary historical observation records or obtained from field campaigns in the target basin. It could be considered a solution to the problem of ungauged basins in some situations. Through application of the SWAT model to four Chinese basins with different climatic and hydrological characteristics, it has been demonstrated that datasets of daily measurements over periods of less than 1 year can constrain simulation uncertainty as effectively as calibration datasets covering several years. In the two wet basins, it was demonstrated surprisingly that the model could be calibrated successfully using only a 1-month dataset, whereas in the two dry basins, longer datasets (6 months) were required and data from wet years and wet periods demonstrated more reliability than data from dry years and dry periods. The results of this study clearly indicate the potential of short-term streamflow observations in calibrating distributed hydrological models for ungauged basins. In real world, it is difficult to assess whether good simulations are achievable with limited calibration data because of the lack of model validation data. Our results show that, the phenomenon that some parameter sets are identified behavioral ones based on the comparison between simulation and observations could be considered as one evidence for making the judgement that the short period data. However, such judgement should be made with careful consideration as our study also shows that it may not be true when the number of the observations is too low or data are observed in a dry year or a dry period. It may only be valid when using data with a length of at least several months and observed in a rainy season or a wet year. To get more general knowledge about when the observations are most informative for model calibration, more researches similar to our studies should be conducted. Based on our findings, the relationship between wetness level of short period data and their effectiveness for parameter calibration are worthy to be explored in this kind of future studies.

## Acknowledgements

This study was supported by the National Natural Science Foundation of China (Grant Nos. 41671018, 91647202), Non-profit Industry Financial Program of Ministry of Water Resources of China (Grant No. 201401036), the National Key Research and Development Program of China (Grant No. 2016YFC0401308) and Fundamental Research Funds for the Central Universities.

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

25

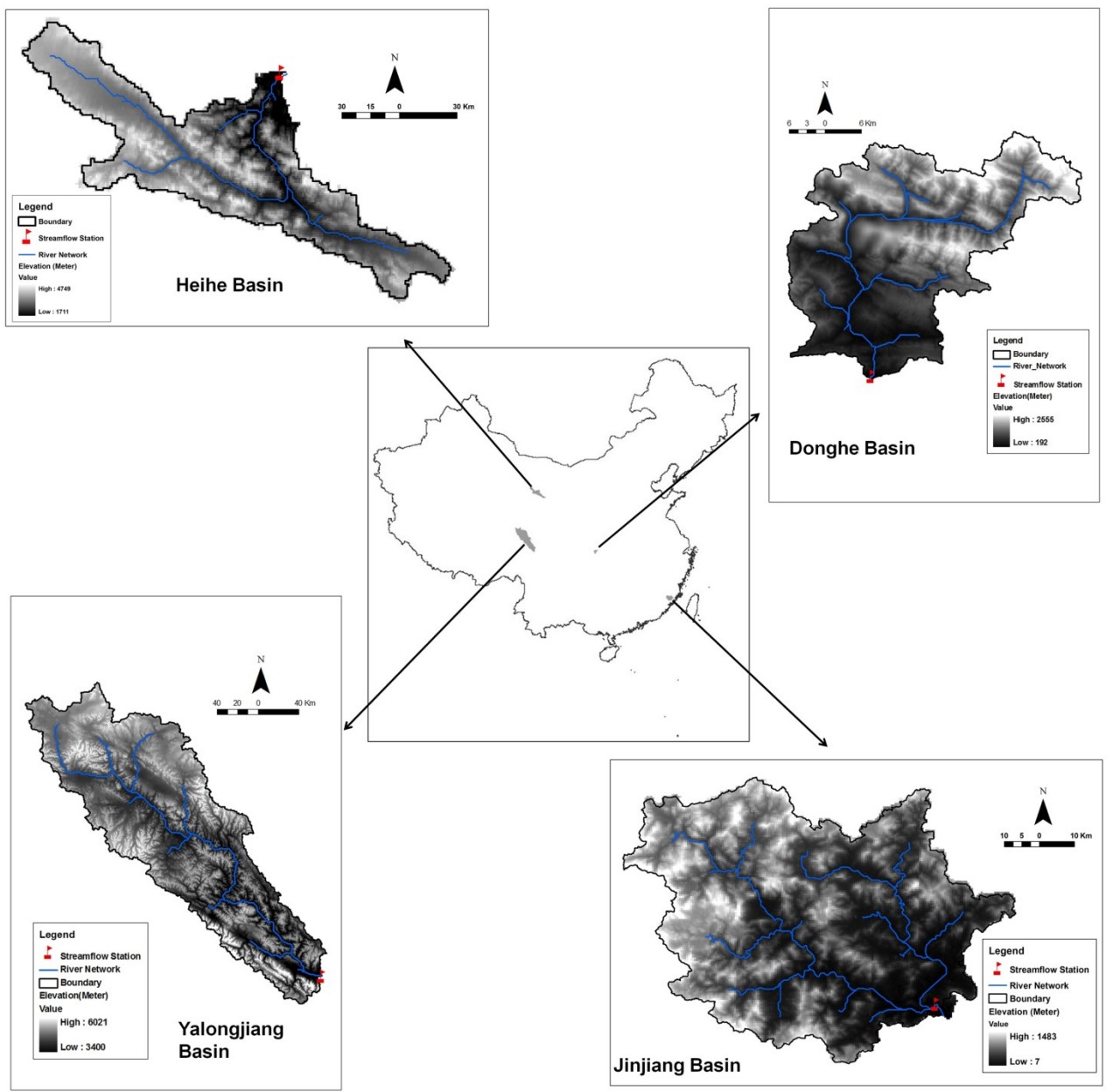

**Figure 1: Topography, river networks and the streamflow gauging stations of the four basins and their locations in China**

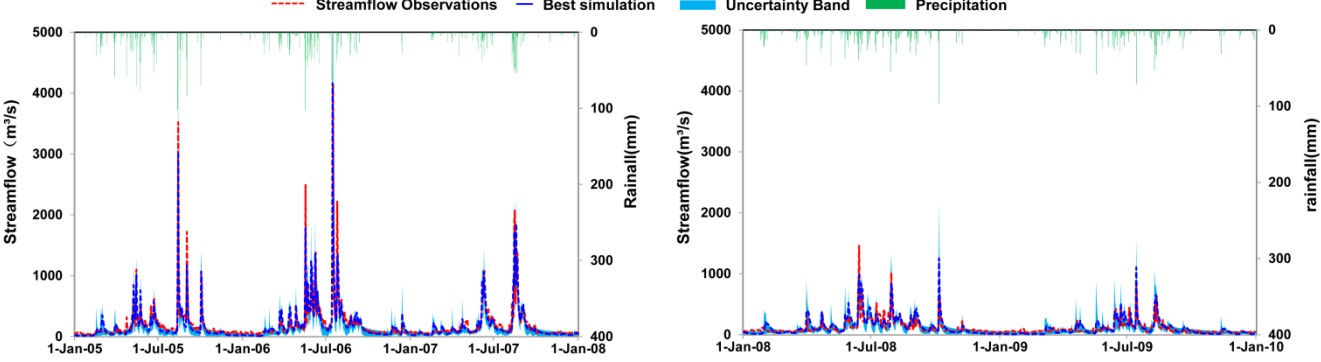

**Figure 2: Simulated streamflow for the benchmark calibration of the Jinjiang Basin in both calibration (2005–2007) and validation period (2008–2009).**

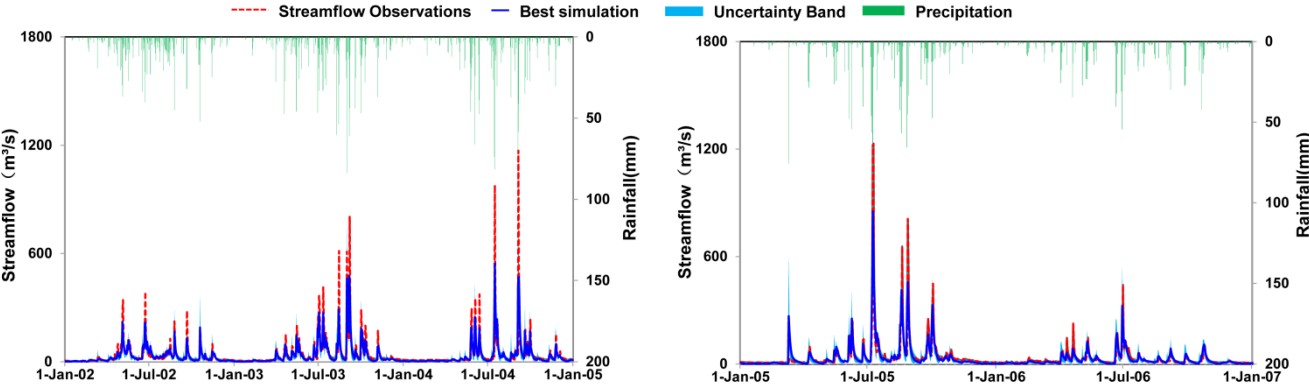

5 **Figure 3: Simulated streamflow for the benchmark calibration of the Donghe Basin in both calibration (2002–2004) and validation period (2005–2006).**

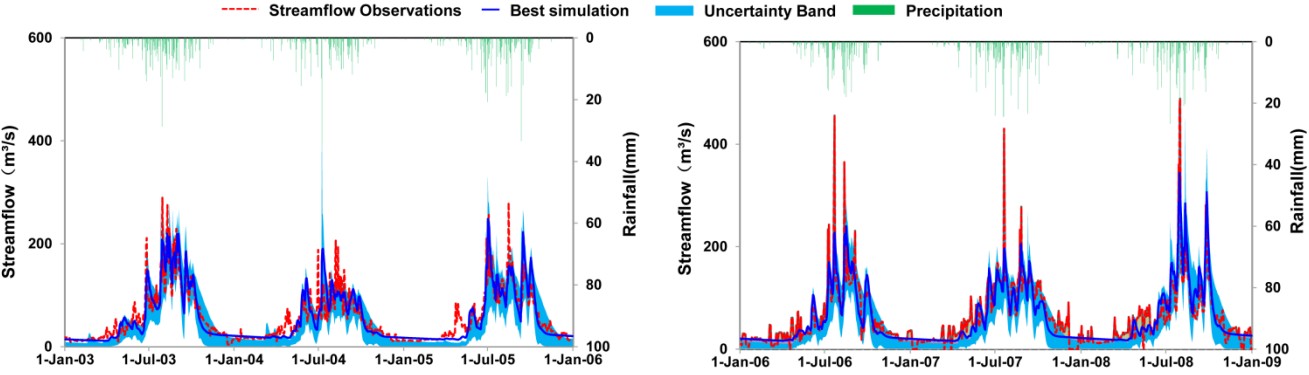

**Figure 4: Simulated streamflow for the benchmark calibration of the Heihe Basin in both calibration (2003–2005) and validation period (2006–2008).**

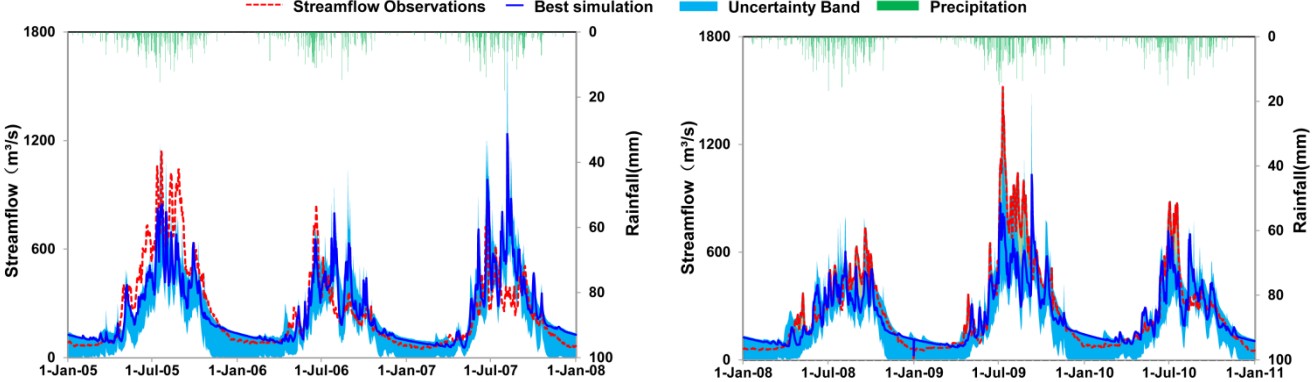

**Figure 5: Simulated streamflow for the benchmark calibration of the Yalongjiang Basin in both calibration (2005–2007) and validation period (2008–2010).**

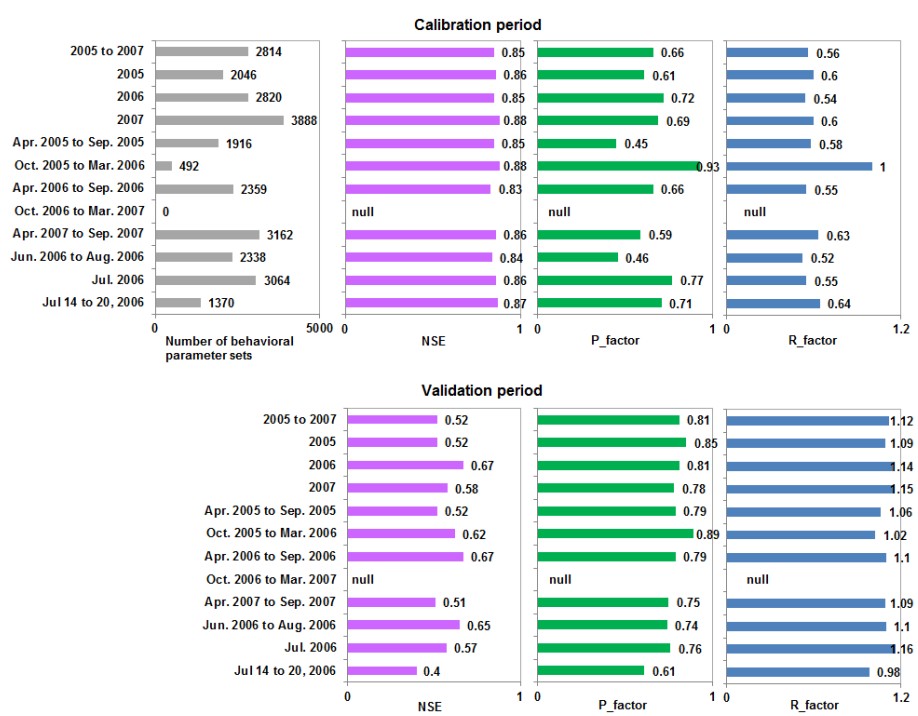

**Figure 6: Model performance for the calibrations using short-period data in Jinjiang Basin**

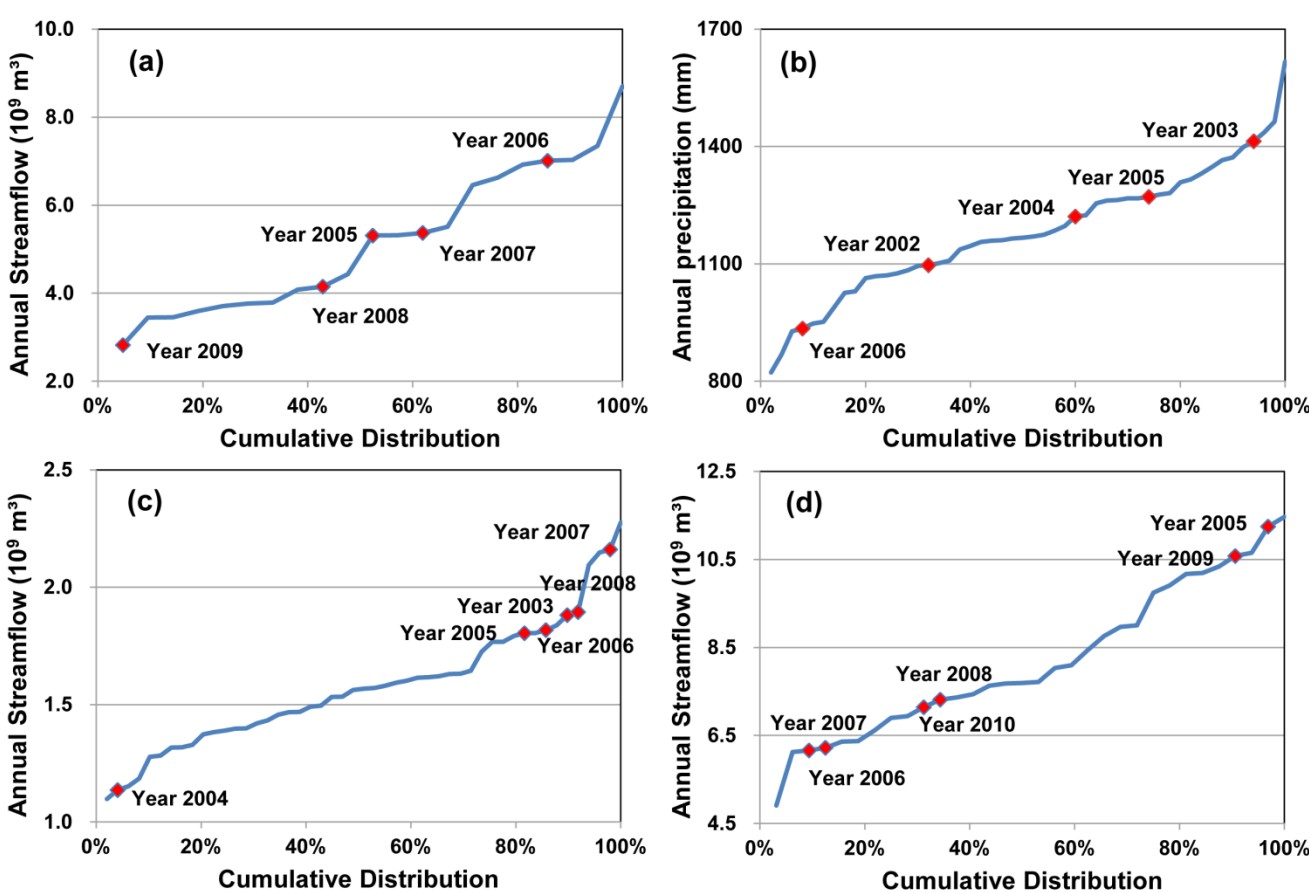

**Figure 7: (a) Cumulative distribution of annual streamflow in Jinjiang Basin (at Shilong station) for the period of 1958 to 2009. (b) Cumulative distribution of annual precipitation in Donghe Basin for the period of 1961 to 2010. (c) Cumulative distribution of annual streamflow in Heihe Basin (at Yingluoxia station) for the period of 1960 to 2008. (d) Cumulative distribution of annual streamflow in Yalongjiang Basin (at Ganzi station) for the period of 1980 to 2011.**

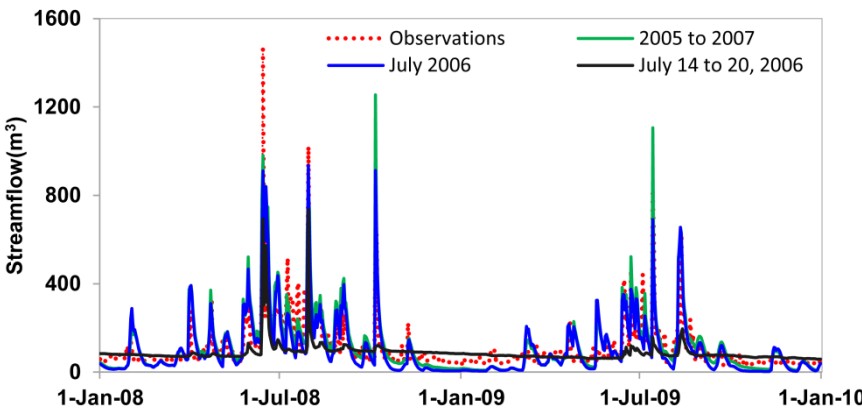

**Figure 8: Simulated streamflow of validation period (2008 to 2009) for the Jinjiang Basin case corresponding to the best performed behavioral parameter set derived from calibration using 3-year data (2005 to 2007), one month(July 2006), one week( July 14 to 20, 2006) and in situ observations.**

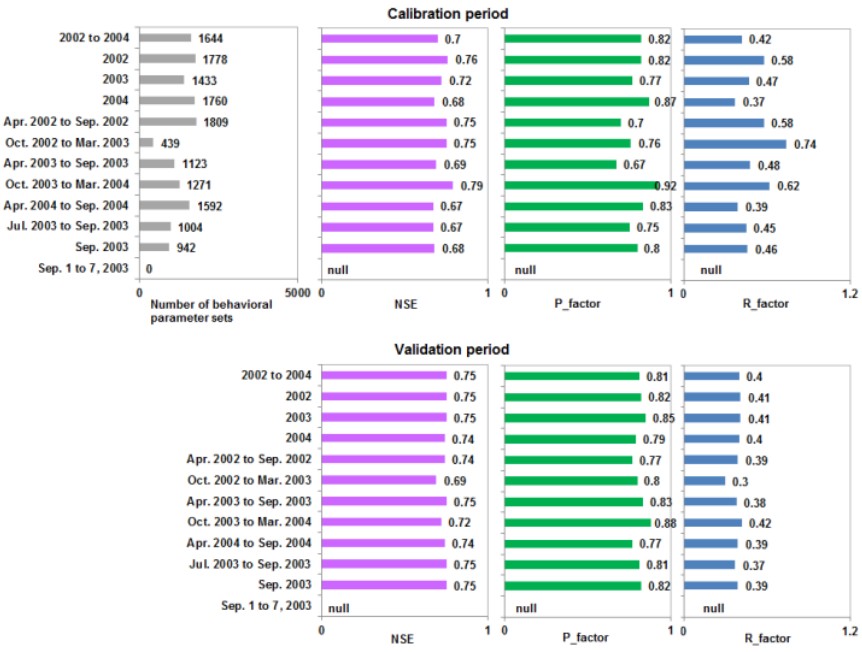

**Figure 9: Model performance for the calibrations using short-period data in Donghe Basin.**

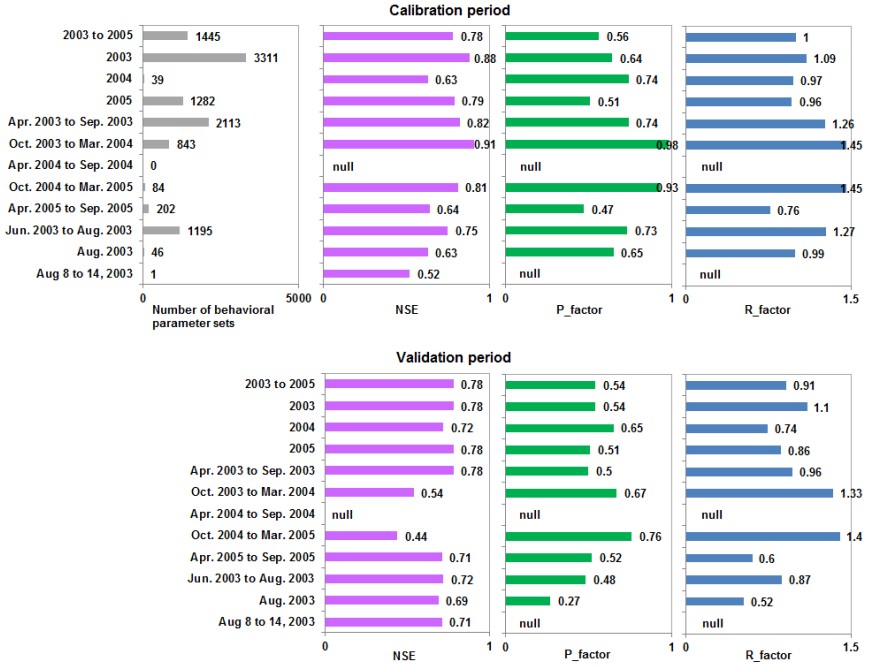

**Figure 10: Model performance for the calibrations using short-period data in Heihe Basin.**

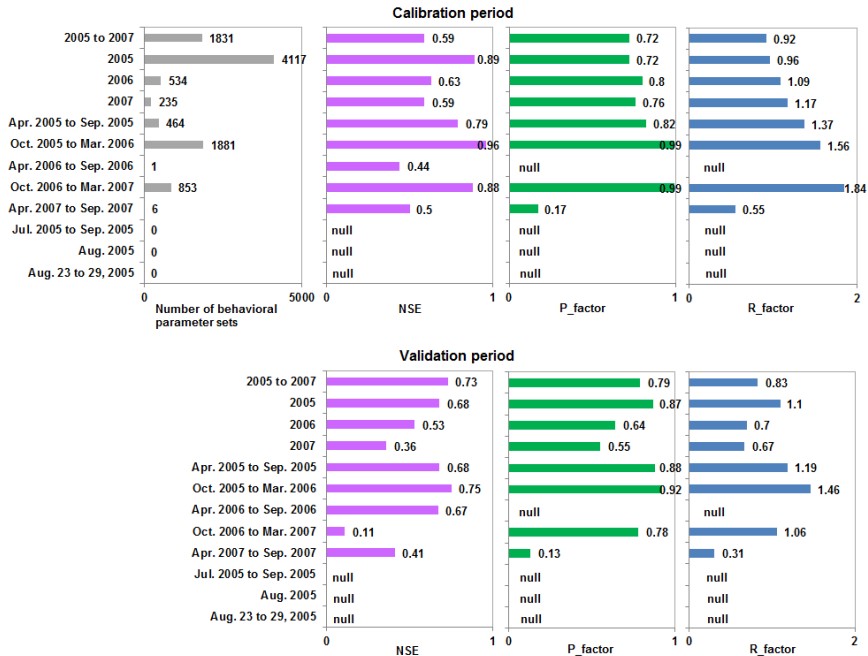

**Figure 11: Model performance for the calibrations using short-period data in Yalongjiang Basin**

**Table 1  Main characteristics of the four basins being studied**

| Basin | Streamflow station | Area (km2) | Climate | Annual Rainfall (mm) | Annual average temperature(℃) | Ranges of Elevation(m) |
|---|---|---|---|---|---|---|
| Jinjiang | Shilong | 5,629 | Subtropical marine monsoon climate | 1651 | 20 | 50 to 1366 |
| Donghe | Wenquan | 1,089 | Subtropical monsoon climate | 1247 | 18 | 192 to 2569 |
| Heihe | Yingluoxia | 8,843 | Continental monsoon climate | 423 | 6 | 1711 to 4749 |
| Yalongjiang | Ganzi | 3,2535 | Continental plateau climate | 570 | 8 | 3400 to 6021 |

**Table 2 The calibration and validation period for the benchmark calibrations of the four basins**

| Basin | Calibration period | Validation period |
|---|---|---|
| Jinjiang | 2005 to 2007 | 2008 to 2009 |
| Donghe | 2002 to 2004 | 2005 to 2006 |
| Heihe | 2003 to 2005 | 2006 to 2008 |
| Yalongjiang | 2005 to 2007 | 2008 to 2010 |

**Table 3 Short periods for which corresponding data were used for the calibrations at the stage one of the evaluation**

| Length of the period | Jinjiang Basin | Donghe Basin | Heihe Basin | Yalongjiang Basin |
|---|---|---|---|---|
| | 2005 | 2002 | 2003 | 2005 |
| One year | 2006 | 2003 | 2004 | 2006 |
| | 2007 | 2004 | 2005 | 2007 |
| | April 2005 to September 2005 | April 2002 to September 2002 | April 2003 to September 2003 | April 2005 to September 2005 |
| | October 2005 to March 2006 | October 2002 to March 2003 | October 2003 to March 2004 | October 2005 to March 2006 |
| Six months | April 2006 to September 2006 | April 2003 to September 2003 | April 2004 to September 2004 | April 2006 to September 2006 |
| | October 2006 to March 2007 | October 2003 to March 2004 | October 2004 to March 2005 | October 2006 to March 2007 |
| | April 2007 to September 2007 | April 2004 to September 2004 | April 2005 to September 2005 | April 2007 to September 2007 |

**Table 4 SWAT model parameters being calibrated**

| Name | Description | Initial range |
|------|-------------|---------------|
| CN2 | SCS runoff curve number | 20–90 |
| EPCO | Plant uptake compensation factor | 0.01–1 |
| GW_DELAY | Groundwater delay time (days) | 30–450 |
| SLSUBBSN | Average slope length (m) | 10–150 |
| ESCO | Soil evaporation compensation coefficient | 0.8–1 |
| ALPHA_BF | Baseflow recession coefficient | 0–1 |
| OV_N | Manning coefficient for overland flow | 0–0.8 |
| CH_K2 | Hydraulic conductivity in main channel (mm/hr) | 5–130 |
| SOL_AWC | Available soil water capacity (mm $H_2O$/mm Soil) | 0–1 |
| SOL_K | Soil Saturated hydraulic conductivity (mm/hr) | 0–2000 |

5    **Table 5 Model performance for the benchmark calibration in the two basins**

| | Number of behavioral parameter sets | NSE | | P_factor | | R_factor | |
|---|---|---|---|---|---|---|---|
| | | Calibration | Validation | Calibration | Validation | Calibration | Validation |
| Jinjiang Basin | 2814 | 0.85 | 0.52 | 0.66 | 0.81 | 0.56 | 1.12 |
| Donghe Basin | 1644 | 0.70 | 0.75 | 0.82 | 0.81 | 0.42 | 0.40 |
| Heihe Basin | 1445 | 0.78 | 0.78 | 0.56 | 0.54 | 1.00 | 0.91 |
| Yalongjiang Basin | 1831 | 0.59 | 0.73 | 0.72 | 0.79 | 0.92 | 0.83 |