# Peer review of "Physically-based distributed hydrological model calibration based on a short period of streamflow data: case studies in four Chinese basins"

_Hydrology and Earth System Sciences, 2016_

## Referee Comment (RC1) · Anonymous Referee #1 · 13 Jun 2016

The paper shows interesting results on distributed hydrological model calibration, in which the authors demonstrate that the SWAT model can be satisfactorily calibrated using 1-6 month daily discharge observation, that is much shorter than normally used for calibration. It can be a large contribution to hydrological modeling for ungauged or poorly gauged basins where long term observation is not available.

There are two comments and recommendations:

1. The major point of this paper is that a hydrological model can be successfully calibrated even based on a short term observation and wet conditions for both period and basin are preferable for effective calibration. It may be true but I wonder if it happens by chance. The authors discuss meteorological conditions of calibration years (2005-2007

for Jinjiang basin and 2003-2005 for Heihe basin) but do not discuss the conditions of validation years. If the study basins were wet in validation years, it is quite reasonable that short observation for wet period can provide successful calibration, while it is truly surprising if it can provide a good result even for the case that validation years are dry. I would like to recommend the authors to add plots for validation years to cumulative distribution shown in Figs 5 and 6 and discuss more about the conditions of validation years in relation to the conditions of calibration periods.

2. I assume that the wet period of the Heihe basin is from April till September and expected that the calibration based on the six months from Apr. 2004 till Sep. 2004 was capable of giving a good result, but Table 5 shows that no behavioral parameter sets were obtained in this period although many behavioral sets were obtained for the two dry periods from 2003 to 2004 and from 2004 to 2005. This is different from the tendency that is found from other calibrations. It would lead to deeper understanding if the authors could give clear explanations for this exceptional case.

---

## Referee Comment (RC2) · Anonymous Referee #2 · 2 Sep 2016

Two watersheds is not enough to conclude your study provides general conclusions! There are groups that use thousands of watersheds, look up large sample hydrology, for example: http://meetingorganizer.copernicus.org/EGU2015/session/18271 http://www.hydrol-earth-syst-sci.net/18/463/2014/hess-18-463-2014.html

The contribution of the paper is not clear. Such analysis on the quality of calibration data dates back to 1996 (http://www.sciencedirect.com/science/article/pii/0022169495029184) and republication in HESS is not justified!

There is no figure provided of how calibration of SWAT with limited data translates into model simulation! How do I know 1 month of data is enough for calibration if I don't see

how the model works graphically? NSE is certainly not enough!

Page 2, lines 1-5: I don't agree with your statement that models like SWAT are able to predict droughts and floods! Droughts and floods respond to climatic forcings and climatic models are used to forecast them, certainly not SWAT!

Page 2, line 8-9: "Most parameters of hydrological models are conceptual without explicit physical meaning, which makes it necessary to identify parameter values through model calibration based on streamflow data". This refers to conceptual models mostly. Physically based distributed are supposed to have parameters with clear physical meaning, that can ideally be measured in the field.

Page 2, Lines 18-20: "Many recent works have focused on using in situ or remote sensing observations of hydrological processes other than streamflow for model calibration, e.g., soil moisture (e.g., Silvestro et al., 2015; Vrugt 20 et al., 2002), evapotranspiration (Vervoort et al., 2014; Winsemius et al., 2008), groundwater level(e.g., Khu et al., 2008)." My understanding is that since streamflow measurements are not available, one can alternatively use other variables such as soil moisture, ground water table and evapotranspiration as calibration data. This is certainly not the case, since measuring these variables is much more difficult and costly than streamflow. I suggest you phrase your sentences more carefully to avoid such confusions.

Page 3, line 4: What do you mean by "changing environment"?

Page 3, lines 4-6: You argue "For hydrological simulations or predictions in changing environments, physically-based distributed hydrological models are usually preferred, because of their better description of the spatial heterogeneity and details of the water cycle at the basin scale (Finger et al, 2012; Wu and Liu, 2012)." I understand that some physical modelers would make such arguments, but it is certainly a debated issue, so I wouldn't make such strong claims. This being said, in a changing climate, even physically based models are not proven to be working properly. The argument made by developers of physically based models is that since they use specific description of the watersheds, their models can handle land-use change (change of physical characteristics of watersheds). This also needs a lot of research still.

Equation 2 is all WRONG! You want to use an objective function of NSE, do, but you can't call it a likelihood function and use it as in the Bayes theorem! There is no scaling in Bayes law! You may call this weight, but not posterior likelihood.

I have a hard time with equation 3 also! Weights (or as you call them posterior likelihoods) are calculate based on overall performance of the model (t=1:N), but are used at each time step to estimate the cumulative probability of streaamflow. This is not right! You want to estimate the 95% uncertainty range, take the 2.5 and 97.5th percentiles of your streamflow simulation ensemble.

Page 6, lines 4-6: I don't understand this sentence, and it is critical to evaluate the methodology proposed in this paper: "As a first trial for the distributed model, we sought to explore the highest possible performance using certain short lengths of records, not the general performance of specific lengths."

The entire section 2.4 is poorly written and methodology is poorly described, making it really difficult to assess the paper.

Page 7, line 2: You admit that it is very time consuming to calibrate SWAT using GLUE. Why not using more intelligent calibration approaches like Markov Chain Monte Carlo? It has been shown in the literature that MCMC is orders of magnitude more efficient than GLUE.

Page 7 line 11: "Kim and Kaluarachchi (2009) and Yapo et al. (1996) showed that data from high-flow periods are more informative than data from low-flow periods for model calibration, because most model modules are activated in high-flow periods." I tend to disagree! It depends on your performance metrics, if you use NSE which is sensitive to peak flows, then yes you are right! But if you use metrics such as baseflow index this is not going to hold. Different processes of a model are activated under different forcings,

so you can't simply ignore several processes and focus on the one (few) process(es) that are activated at the wet condition.

Page 7 line 24: Uncertainty bound not band! Correct throughout the manuscript.

Page 8, lines 2-3: "For the 1-year period, all three calibrations performed similarly to the benchmark calibration, and the dataset for 2006 even outperformed the benchmark" It is interesting and concerning that a shorter calibration period provides a higher performance. It requires explanation as to how it happened! You can't just leave it like that which might spuriously suggest smaller calibration period is sometimes even better! Here are my thoughts: 1. You are not using a consistent period to evaluate your model! 2. Your calibration approach did not converge to the right posterior distribution (as might happen with GLUE) 3. Your data includes some misinformation, meaning not only it doesn't provide any good information to constrain model parameters, but also it misguides the model! In the cases of 1 & 2, extra data can only be redundant and cannot deteriorate the performance of the model!

Page 8, lines 14-16: "The calibration using the 1-month dataset still achieved similar performance to benchmark calibration. Thus, it is indicated that in the Jinjiang Basin, it is possible to calibrate the SWAT model effectively using only 1-month's continuous daily observations of streamflow." This claim is rather strange to me! One month is enough to capture all the processes? Some processes might not even be activated in one month! Again, this is because you focused all your attention on NSE, and what is most important in NSE is the high peaks. So if you activate the processes that reproduce the high peaks, you get a good performance. This doesn't mean one month is enough to calibrate a model!

---

## Author Comment (AC1) · 22 Sep 2016

**Dear reviewer,**

**Thank you very much for your comments and recommendations. Our responses to the two important comments are as follows. If you feel more explanation is needed. Please do not hesitate to contact with us.**

**From the authors**

*The paper shows interesting results on distributed hydrological model calibration, in which the authors demonstrate that the SWAT model can be satisfactorily calibrated using 1-6 month daily discharge observation, that is much shorter than normally used for calibration. It can be a large contribution to hydrological modeling for ungauged or poorly gauged basins where long term observation is not available.*

*There are two comments and recommendations:*
*1. The major point of this paper is that a hydrological model can be successfully calibrated even based on a short term observation and wet conditions for both period and basin are preferable for effective calibration. It may be true but I wonder if it happens by chance. The authors discuss meteorological conditions of calibration years (2005-2007 for Jinjiang basin and 2003-2005 for Heihe basin) but do not discuss the conditions of validation years. If the study basins were wet in validation years, it is quite reasonable that short observation for wet period can provide successful calibration, while it is truly surprising if it can provide a good result even for the case that validation years are dry. I would like to recommend the authors to add plots for validation years to cumulative distribution shown in Figs 5 and 6 and discuss more about the conditions of validation years in relation to the conditions of calibration periods.*

**Response:**

The validation years are also added to the cumulative distribution as shown in Figure R1 and R2. For the Heihe Basin (Figure R1), the validation years (2006, 2007 and 2008) are all wet years. Generally, as shown in many studies, model performance in the validation period will decrease, compared with calibration period. However, for the benchmark calibration of the Heihe basin, the performance between validation and calibration period does not show much difference, as shown in Table 3 of the manuscript. This phenomenon may come from the fact revealed by Figure R1 that, the three validation years (2006, 2007, and 2008) are all wet years. And two of the three calibration years (2003, 2005) are also wet years. Calibration data may contains sufficient information for parameters identification in wet years.

[Figure]

Figure R1 Cumulative distribution of annual streamflow at Yingluoxia station for the period of 1960 to 2008

For the benchmark calibration in the Jinjiang Basin, compared with calibration period, model performance in the validation period decreases. Figure R2 revealed that the two validation years (2008, 2009) are dry years. The three calibrations years are average (2005) and wet year (2006, 2007). The decrease in model performance is consistent with other studies (e.g. Todorovic and Plavsic 2016), in which model efficiency also decrease when calibration period is wetter than validation period.

[Figure]

Figure R2 Cumulative distribution of available annual streamflow at Shilong station for the period of 1958 to 2009

***2. I assume that the wet period of the Heihe basin is from April till September and expected that the calibration based on the six months from Apr. 2004 till Sep. 2004 was capable of giving a good result, but Table 5 shows that no behavioral parameter sets were obtained in this period although many behavioral sets were obtained for the two dry periods from 2003 to 2004 and from 2004 to 2005. This is different from the tendency that is found from other calibrations. It would lead to deeper understanding if the authors could give clear explanations for this exceptional case.***

**Response:**

The Heihe basin is an inland basin in the arid northwest China. Most rainfall occurs in the summer season. About 75% of total annual streamflow come from the wet period from April to

September. As shown in Figure R1, 2004 is an extremely dry year. Compared with normal year, either in wet season or in dry season of 2004, the average streamflow decreased. Considering the big contribution to total annual streamflow, the degree of decreases of streamflow in the wet period has high possibility to be bigger than dry season. The runoff generation mechanism in this wet season with extremely low streamflow is very different from normal seasons, which made the model cannot capture the essence of variation in streamflow, therefore none of the randomly generated 10,000 parameter sets can reproduce the hydrograph of this wet season with acceptable accuracy. This is our explanation about why no parameter set was identified as behavioral ones using data of wet season in 2004.

**References:**
Todorovic, A. and Plavsic, J.: The role of conceptual hydrologic model calibration in climate change impact on water resources assessment, J. Water Clim. Change, 7 (1), 16-28, 2016

---

## Author Comment (AC2) · 22 Sep 2016

**Dear reviewer,**

**Thank you very much for your constructive comments which help us to clarify the contribution of our study and describe the knowledge gained from this study more precisely. All questions being addressed will be answered in details. If you feel it is still insufficient, please do not hesitate to contact with us. We would like to make further explanations and revisions.**

**From the authors**

*Two watersheds is not enough to conclude your study provides general conclusions! There are groups that use thousands of watersheds, look up large sample hydrology, for example: http://meetingorganizer.copernicus.org/EGU2015/session/18271*
*http://www.hydrol-earth-syst-sci.net/18/463/2014/hess-18-463-2014.html*

**Responses:**

The researches mentioned by the reviewer are expected to get general knowledge in many situations by applying hydrological model in large number of basins. The objective of our study is to demonstrate short periods of observations data have the possibility to be useful for physically-based hydrological model calibration in data-sparse basin. The paper could inspire more researchers to think about using such dataset to calibrate distributed hydrological models in basins lacking of streamflow data. In the past, this approach didn't draw much attention for solving the calibration problem of distributed model in ungauged basins. We focus the potential of this method, rather than the general applicability. Based on the objective of this study, two basins with very different climatic and geophysical conditions were selected to conduct our study.

*The contribution of the paper is not clear. Such analysis on the quality of calibration data dates back to 1996 (http://www.sciencedirect.com/science/article/pii/0022169495029184) and republication in HESS is not justified!*

**Response:**

The model used in Yapo *et al.*(1996) is a conceptual rainfall-runoff model. In comparison, our study focuses on *physically-based distributed hydrological models*. This is one major difference between our study and Yapo *et al.* (1996). These two types of models face the same problem: lacking of streamflow data for calibrations in ungauged basins. For conceptual rainfall-runoff models, effectiveness of using short period of streamflow data to calibrate the model has been demonstrated in several papers, such as the one Yapo et al.(1996) mentioned by the reviewer, also Perrin et al.(2006), Beven and Tada (2012). However, based on our knowledge, such approach has not been widely tested for physically-based distributed hydrological models. The contribution of this paper is to demonstrate that, using short period of streamflow data also have the possibility to be useful for calibrating physically-based distributed hydrological models, which are usually preferred, because of their better description of the spatial heterogeneity and details of the water cycle at the basin scale.

*There is no figure provided of how calibration of SWAT with limited data translates into model simulation! How do I know 1 month of data is enough for calibration if I don't see how the*

*model works graphically? NSE is certainly not enough!*

**Response:**

Thank you for this suggestion. For the Jinjiang Basin, a figure showing the simulated hydrograph corresponding to the calibration using data of one month will be added to the revised manuscript to demonstrate the effectiveness of using such short period of data for calibration. The figure is also shown here (Figure R3 below). For the validation period (2008 to 2009), it is shown that the best simulations of ensemble predictions corresponding to the calibrations using three year data (2005 to 2007) and one month data (July 2006) is quite similar visually. As a response to the concern about NES, we computed the Mean Absolute Error (MAE) of best simulations in low flow period (September to next March) and high flow period (April to September). In high flow period, the MAE of the best simulation corresponding to calibration using three-year data and one-month-data is 66.3 $m^3$/s and 63.3$m^3$/s, respectively. In low flow period, the MAE of the two best simulations is 36.4$m^3$/s and 43.1$m^3$/s, respectively. Generally, similar performance level is achieved by the two best simulations. These results could support our conclusion that in the Jinjiang Basin, it is possible that one month data is informative to calibrate the SWAT model effectively.

[Figure]

Figure R3 Observed streamflow of 2008 to 2009(dashed red line), best simulations of ensemble prediction for the validation period (2008-2009) corresponding to calibration based on streamflow data of 2005 to 2007 (solid black line) and July 2006(solid green line) in the Jinjiang Basin.

*Page 2, lines 1-5: I don't agree with your statement that models like SWAT are able to predict droughts and floods! Droughts and floods respond to climatic forcings and climatic models are used to forecast them, certainly not SWAT!*

**Response:**

As mentioned by the reviewer, hydrological models, like SWAT, use climatic data as forcing data to predict streamflow in the river. When climatic forecasting are available, hydrological models can employ it as input and predict hydrological drought and flood, from the perspective of quantity of streamflow.

*Page 2, line 8-9: "Most parameters of hydrological models are conceptual without explicit physical meaning, which makes it necessary to identify parameter values through model calibration based on streamflow data". This refers to conceptual models mostly. Physically*

*based distributed are supposed to have parameters with clear physical meaning, that can ideally be measured in the field.*

**Response:**

Although values of parameters with explicit physical meaning can be measured, the scale of measurement and model simulation is different, which makes it difficult to apply measured values to hydrological models directly. Also, such measurements require intensive field survey, which are not available in most researches. Therefore, usually model parameters are obtained from model calibration based on streamflow data. Such understanding can be found in published literatures (e.g., Gupta et al. 2005).

*"Many recent works have focused on using in situ or remote sensing observations of hydrological processes other than streamflow for model calibration, e.g., soil moisture (e.g., Silvestro et al., 2015; Vrugt 20 et al., 2002), evapotranspiration (Vervoort et al., 2014; Winsemius et al., 2008), groundwater level(e.g., Khu et al.,2008)." My understanding is that since streamflow measurements are not available, one can alternatively use other variables such as soil moisture, ground water table and evapotranspiration as calibration data. This is certainly not the case, since measuring these variables is much more difficult and costly than streamflow. I suggest you phrase your sentences more carefully to avoid such confusions.*

**Response:**

Our intention is not to make new observations of other variables of hydrological cycles, but to make best use of available data of such variables. There are cases that in some basins, streamflow data is unavailable, but measurements of the other variables are available. In such situation, the available data maybe valuable for model calibration. To avoid confusions, these sentences will be revised based on above understanding.

*Page 3, line 4: What do you mean by "changing environment"?*

**Response:**

This term of "changing environment" comes from the paper of Montanari et al. (2013), which introduces the IAHS Scientific Decade 2013–2022 " Panta Rhei—Everything Flows". It means the all factors that influence hydrological system, including both changes in nature (e.g., climate changes) and society (e.g., global population growth).

*Page 3, lines 4-6: You argue "For hydrological simulations or predictions in changing environments, physically-based distributed hydrological models are usually preferred, because of their better description of the spatial heterogeneity and details of the water cycle at the basin scale (Finger et al, 2012; Wu and Liu, 2012)." I understand that some physical modelers would make such arguments, but it is certainly a debated issue, so I wouldn't make such strong claims. This being said, in a changing climate, even physically based models are not proven to be working properly. The argument made by developers of physically based models is that since they use specific description of the watersheds, their models can handle land-use change (change of physical characteristics of watersheds). This also needs a lot of research still.*

**Response:**

We agree with the reviewer's opinion about the comparison between conceptual and physically based hydrological model. The sentences will be revised in the new version of manuscript to

express our understanding more precisely.

*Equation 2 is all WRONG! You want to use an objective function of NSE, do, but you can't call it a likelihood function and use it as in the Bayes theorem! There is no scaling in Bayes law! You may call this weight, but not posterior likelihood.*

**Response:**

For GLUE, the term of likelihood is used in a very general sense, as described by developer of GLUE, Keith Beven, in the paper of Beven and Binley (1992): the likelihood function in GLUE works as a fuzzy, belief, or possibilistic measure of how well the model conforms to the observed behavior of the system, and not in the restricted sense of maximum likelihood theory. The likelihood measure quantifies the difference between simulation and observations. The only requirement for a likelihood measure is that it should be assigned as zero for all parameter sets that cannot reproduce the observations and should increase monotonically as the performance rises. Based on the theory of GLUE, in our opinion, using NSE as a likelihood function is proper in our study. Also the Bayes equation is employed in GLUE in a general sense. Equation 2 has been used in many studies related to GLUE, such as Freer and Beven (1996), Beven and Freer (2001).

*I have a hard time with equation 3 also! Weights (or as you call them posterior likelihoods) are calculate based on overall performance of the model (t=1:N), but are used at each time step to estimate the cumulative probability of streaamflow. This is not right! You want to estimate the 95% uncertainty range, take the 2.5 and 97.5th percentiles of your streamflow simulation ensemble.*

**Response:**

For application of GLUE, the posterior likelihood is computed based on the overall model performance in the period when observations are available to compare model behavior with observations. Then at each time step, the posterior likelihood is used to estimate cumulative probability of streamflow. This is the way how GLUE estimates simulation uncertainty, and equation 3 can be found in many studies using GLUE. In our opinion, this equation is correct.

*Page 6, lines 4-6: I don't understand this sentence, and it is critical to evaluate the methodology proposed in this paper: "As a first trial for the distributed model, we sought to explore the highest possible performance using certain short lengths of records, not the general performance of specific lengths."*

**Response:**

As a first trial for distributed model calibration using limited observations, our objective is to show the possibility of using short period of data for calibration, not to identify the exact minimum requirement for the length of calibration data. Therefore, what we concern is the best performance when using certain short length period of records, not the general performance when using different datasets of certain short length period. This explanation will be added to the revised version of paper.

*The entire section 2.4 is poorly written and methodology is poorly described, making it really difficult to assess the paper.*

**Response**:

This section will be revised in the new manuscript, to make it easier to be understood as follows:

For the two basins, firstly we carried out benchmark calibration using three year daily observations for model calibration: For the Jinjiang Basin, the calibration period is 2005-2007 and the validation period is 2008 and 2009. For the Heihe Basin, the calibration period is 2003-2005 and the validation period of is 2006-2008. Secondly, in order to test whether using short period of observations could calibrate the model effectively (i.e., achieve similar performance as benchmark calibration), subsets of the data used for model calibration in the two benchmark calibrations will be selected and used for model calibration. Then, the results of these calibrations (i.e., performance of the calibrated model) will be compared with the benchmark calibration of each basin. If the performance is similar to that of benchmark calibration, it will be conclude that it is possible that data of that specific short period is as informative as three year observations (i.e., data used in benchmark calibration) for parameter calibration and then lead to the conclusion that for calibrating physically based distributed hydrological model in data-sparse basin, resorting to calibration using short period of observations is a possible way.

For all calibrations of each basin, the calibration period is either three-year observations (benchmark calibration) or a subset of the three-year observations (calibrations using short periods of data). But the validation period of all calibrations are made same (2008-2009 for Jinjiang Basin; 2006-2008 for the Heihe Basin), to ease the comparison between benchmark calibration and calibration using short period of data. In such contexts, two issues are extremely important to achieve the goal of this study, the method for assessment of calibrated model performances, and the strategy of selecting short periods of data from streamflow observations used in benchmark calibrations. The details about the two issues are briefly introduced as follows:

The evaluation of each calibration was performed from the aspects of general performance and simulation uncertainty. The general performance was represented by the NSE of the best behavioral parameters set (i.e., the one with the highest likelihood value constrained by the calibration data). The simulation uncertainty is quantified by an index named as "U", which combine the percentage of observations covered by the uncertainty band and the average width of the uncertainty band. The definition of U can be found in the original manuscript. The NSE and U are computed for the calibration and validation period for all model calibration. For the evaluation, we focus on values of NSE and U for validation period, as information in this period was not used for model calibration. The validation period for all calibrations in one specific basin is made to be same (2006-2008 for the Heihe Basin; 2008-2009 for the Jinjiang Basin.), for conducting the comparison among calibrations in an objective manner.

For extracting subset from the three-year data used in the benchmark calibrations, it was impossible to follow the studies of conceptual models that could conduct calibrations many times, due to the high demand of time for distributed model simulation. We have to perform the calibration in manageable times. In such situation, how to select subset from the three-year dataset is important. Three calibrations using 1-year data record that covered both the rainy and dry seasons, and five calibrations using 6-month data record that covered either a rainy season or a dry season were undertaken. Many studies of conceptual model showed that, generally, when the number of observations for model calibration is same, the data for high flow period are most informative for model calibration. Based on this general understanding, the 3-month, 1-month and 1-week period with highest average streamflow in the best performed 6-month dataset were selected as the representative dataset for the above mentioned temporal scale. Then these

subsets were used for model calibration.

By such arrangement, the possibility of using 1-year, 6-month, 3-month, 1-month and 1-week dataset for model calibration were evaluated from the aspects of general performance and simulation uncertainty. Meanwhile, the total time needed for all model calibration is acceptable.

*Page 7, line 2: You admit that it is very time consuming to calibrate SWAT using GLUE. Why not using more intelligent calibration approaches like Markov Chain Monte Carlo? It has been shown in the literature that MCMC is orders of magnitude more efficient than GLUE.*

**Response:**

For daily-step model simulation of several years, it is common for lumped conceptual models to finish the simulation within one second. However, for distributed model like SWAT, usually, one such model run may need several minutes. We have no information about prior distribution of model parameters, based on which, parameter sets will be generated randomly. The uniform distribution is used as the prior distribution of each parameter. In such situation, implementing GLUE with Latin hypercube sampling is usually preferred. This strategy has been used in many literatures related to GLUE and SWAT simulation.

*Page 7 line 11: "Kim and Kaluarachchi (2009) and Yapo et al. (1996) showed that data from high-flow periods are more informative than data from low-flow periods for model calibration, because most model modules are activated in high-flow periods." I tend to disagree! It depends on your performance metrics, if you use NSE which is sensitive to peak flows, then yes you are right! But if you use metrics such as baseflow index this is not going to hold. Different processes of a model are activated under different forcings, so you can't simply ignore several processes and focus on the one (few) process(es) that are activated at the wet condition.*

Response:

We agree with the reviewer that the statement of "because most model modules are activated in high-flow period" is improper and it will be deleted from the manuscript.

*Page 7 line 24: Uncertainty bound not band! Correct throughout the manuscript.*

**Response:**

Uncertainty bounds are two lines consisted of the 2.5% and 97.5% quantiles of simulated streamflow in each time step. The uncertainty band is the area bounded by these two lines in hydrograph. Both term of "uncertainty bound" and "uncertainty band" (e.g., Beven and Benley, 1992; Yang et al., 2008) are used when applying GLUE for uncertainty analysis.

*Page 8, lines 2-3: "For the 1-year period, all three calibrations performed similarly to the benchmark calibration, and the dataset for 2006 even outperformed the benchmark" It is interesting and concerning that a shorter calibration period provides a higher performance. It requires explanation as to how it happened! You can't just leave it like that which might spuriously suggest smaller calibration period is sometimes even better! Here are my thoughts: 1. You are not using a consistent period to evaluate your model! 2. Your calibration approach did not converge to the right posterior distribution (as might happen with GLUE) 3. Your data includes some misinformation, meaning not only it doesn't provide any good information to constrain model parameters, but also it misguides the model! In the cases of 1 & 2, extra data*

*can only be redundant and cannot deteriorate the performance of the model!*

**Response:**

The evaluation of each calibration is based on judging model performance in the validation period. The validation period is made to be same for all calibrations in each basin. For the Jinjiang Basin, the validation period is 2008 to 2009. For the Heihe Basin, the validation period is 2006 to 2008. Streamflow data are obtained from the water administrative department in local government and have used in many studies. The data quality is guaranteed. We apologize that the statement of "the dataset for 2006 even outperformed the benchmark" is misleading. It is only based on the NSE of best performed behavioral parameter set. Another important aspect of the evaluation is simulation uncertainty. It is quantified by the index of "U" as defined in the manuscript. As shown in the Table 4 of the original manuscript, it is indicated that the simulation uncertainty of calibration using data of 2006 is a little bit higher than benchmark calibration. So we agree that the statement is improper and it is not our intention to conclude that using shorter period of calibration is better than using long period of observations. The method presented in this study is only expected to be useful for model calibration in data-sparse basins where streamflow data of several years are unavailable. Therefore the statement of "the dataset for 2006 even outperformed the benchmark" will be deleted.

*Page 8, lines 14-16: "The calibration using the 1-month dataset still achieved similar performance to benchmark calibration. Thus, it is indicated that in the Jinjiang Basin, it is possible to calibrate the SWAT model effectively using only 1-month's continuous daily observations of streamflow." This claim is rather strange to me! One month is enough to capture all the processes? Some processes might not even be activated in one month! Again, this is because you focused all your attention on NSE, and what is most important in NSE is the high peaks. So if you activate the processes that reproduce the high peaks, you get a good performance. This doesn't mean one month is enough to calibrate a model!*

**Response:**

We apologize that we didn't give enough details for the evaluations of calibration using 1-month data in the manuscript. As our responses to previous comments, from simulated hydrograph and all model performance indexes (NSE, U, MAE for both low and high flow period) in the validation period, it is indicated that the calibrated model corresponding to 1-month calibration data performs similar to the benchmark calibration.

We realize that our expression about the results using 1-month data is too strong and confusing. In the revised manuscript, instead of the original expression, we will conclude that it is possible that 1-month's continuous daily observations can contain much of the information content of 3-year continuous streamflow data for model calibration.

**References:**

Beven, K. and Binley, A.: The future of distributed models: Model calibration and uncertainty prediction, Hydrol. Process., 6, 279-298, 1992.

Beven, K. and Freer, J.: Equifinality, data assimilation, and uncertainty estimation in mechanistic modelling of complex environmental systems using the GLUE methodology, J. Hydrol., 249, 11-29, 2001.

Freer, J. and Beven, K., Bayesian estimation of uncertainty in runoff predication and the value of

data: An application of the GLUE approach, Water Resour. Res. 32, 2161-2173. 1996.

Gupta, H. V. ,Beven, K. J. and Wagener, T.: Model Calibration and Uncertainty Estimation, In Encyclopedia of hydrological science, Anderson MG (eds), John Wiley & Sons, Ltd,2006.

Montanari, A., Young, G., Savenije, H.H.G., Hughes, D., Wagener, T., Ren, LL., Koutsoyiannis, D., Cudennec, C., Toth, E., Grimaldi, S., Blöschl, G., Sivapalan, M., Beven, K., Gupta, H., Hipsey, M., Schaefli, B., Arheimer, B., Boegh, E., Schymanski, S.J., Baldassarre, G.D., Yu, B., Hubert, P., Huang, Y., Schumann, A., Post, D.A., Srinivasan, V., Harman, C., Thompson, S., Rogger, M., Viglione, A., McMillan, H., Characklis, G., Pang, Z., Belyaev, V.: "PantaRhei—Everything Flows": Change in hydrology and society—The IAHS Scientific Decade 2013-2022. Hydrolog. Sci. J., 58(6),1256-1275, 2013.

Perrin, C., Oudin, L., Andreassian, V., Rojas-Serna, C., Michel, C., and Mathevet, T.: Impact of limited streamflow data on 30 the efficiency and the parameters of rainfall-runoff models, Hydrolog. Sci. J., 52, 131–151, 2007.

Tada, T. and Beven, K. J.: Hydrological model calibration using a short period of observations, Hydrol. Process., 26, 883-892, 2012.

Yang, J., Reichert, P., Abbaspour, K. C., Xia, J. and Yang, H.: Comparing uncertainty analysis techniques for a SWAT application to the Chaohe Basin in China, J. Hydrol., 358, 1-23, 2008.

Yapo, P. O. and Gupta, H. V. and Sorooshian, S.: Automatic calibration of conceptual rainfall-runoff models: sensitivity to calibration data, J. Hydrol., 181, 23-48, 1996.

---

## Author Response (AR1)

Dear editor and reviewers,

Thank you very much for your constructive comments and recommendations. Our responses to comments, list of changes and the marked-up manuscript are as follows. If you feel more explanations or revisions are needed. Please do not hesitate to contact with us.

Best regards,

From the authors

**Responses to the editor:**

*(1) As is suggested by reviewer #2, two watersheds, typical though, are far from enough. If this manuscript is designed to only provide an idea or assumption on "short period calibration of hydrological models", it does not contribute a lot to the hydrological community as many previous researches have already done this using different models though (same comments as reviewer #2). Therefore, I think you may conduct few more case studies, test it and make it convincing. Or if this manuscript focuses on short period calibration of "Physically-based distributed hydrological model", a discussion on the calibration of both conceptual models and physical models is expected. And also their differences in calibration should be discussed. In addition, what can we learn from this idea and how can we use this idea in the calibration of hydrological models for data-scarce basins?*

**Response:**

To make the findings of this study more convincing, we added two more basins to our study: the upstream Yalongjiang Basin with Ganzi station as the outlet, which is a dry basin in the Qinghai-Tibet Plateau, and the Donghe Basin, a wet basin in the upstream region of Three Gorges Reservoir. By doing this, we could test our idea in two wet basins, Jinjiang Basin and Donghe Basin, and two dry basins, upstream Heihe Basin and upstream Yalongjiang Basin, to improve the universality of our conclusions. The main findings of this study are as follows:

1. In the four basins, it is possible that data records with lengths of less than one year could calibrate the model effectively. The models in the two wet basins could be calibrated effectively using a shorter period of records (one month) than the two arid basins (six months).

2. When using one-year data for calibration, for the two wet basins, whether the year is wet or dry have little influences on model calibration. However, in the two dry basins, the influence is significant: data of dry years are less reliable than the ones of wet years.

3. When using 6-month observations for calibration, in all of the four basins the model performances are more sensitive to the timing of the observations than using 1-year data. And this sensitivity is higher for the two dry basins. Data from wet year or wet season are more reliable than the ones from dry year or dry season, especially for the two dry basins.

From these findings, in our opinion, the contributions of this study to utilization of short periods of observations for calibrating physically based distributed hydrological model are as follows:

The paper could inspire more researchers to think about using such dataset to calibrate distributed hydrological models in basins lacking of streamflow data. Many past researches concentrate on this approach for the calibration of lumped conceptual model. However, the possibility for the physically based distributed hydrological model, like SWAT used in this study, has not been discussed widely. Our study confirms that for physically based distributed hydrological models, this approach could also work well.

More importantly, this study is valuable for evaluating the effectiveness of short period data for model calibration in real world application. Our results show that, the phenomenon that some parameter sets are identified behavioral ones based on the comparison between simulation and observations could be considered as one evidence for making the judgement that the short period data. However, such judgement should be made with careful consideration as our study also shows that it may not be true when the number of the observations is too low or data are observed in dry years or dry period. It may only be valid when using data with a length of several months and observed in rainy season or wet years. To get more general knowledge about when the observations are most informative for model calibration, more researches similar to our studies should be conducted. Based on our

findings, the relationship between wetness level of short period data (i.e., the records were observed in a wet year or dry year and in rainy season or dry season) and their effectiveness for parameter calibration are worthy to be explored in this kind of future studies.

The manuscript has been revised accordingly.

*(2) In your responses to reviewer #1, as is concluded that the good model performance for the validation period "may come from the fact that, the three validation years (2006, 2007, and 2008) are all wet years. And two of the three calibration years (2003, 2005) are also wet years. Calibration data may contain sufficient information for parameters identification in wet years." If in this case, the model is useful only when the calibration period and validation period both are wet years or both are dry years. What can we do when only one year's data is available for a data-scarce watershed? We cannot tell if this year is as the same condition as the following/previous years. Therefore, the objective of this study "how the use of limited continuous daily streamflow data might support the application of a physically-based distributed model in data-sparse basins" is not well explained.*

**Response:**

With the two new cases, we have gained deeper insight about the questions addressed by the editor and reviewer #1. In the two wet basins, either using 1-year data of wet year or dry year could achieve performance similar to calibration using three year data in the validation period, which contains both dry years and wet years in the new Donghe Basin case and contains two dry years in the Jinjiang Basin case. For the two dry basins, one wet year data could also work well in validation period (also covers both wet years and dry years). However, one dry year data performs worse than data of wet years. Especially, for the case of Yanlongjiang Basin, model performances of validation period corresponding to extreme dry year 2006, 2007 deviate from calibration using three-year data to an unacceptable extent, i.e. fail to reproduce streamflow in validation period. In summary, one wet year data are useful to reproduce streamflow in both dry and wet basins. One dry year data in wet basins also can works well. But one dry year data in dry basin have the high possibility that it cannot calibrate the model effectively. In such context, to know whether the year is a wet year or a dry year from the annual streamflow frequency is useful to judge the effectiveness of 1-year calibration data. In the case of only one year data is available, it is impossible to obtain yearly streamflow frequency directly. Precipitation data are much easier to be obtained than streamflow data, either from in situ gauging or remote sensing data. Nowadays, many precipitation data product with wide spatial and temporal coverage are available, from which precipitation frequency can be computed. As streamflow is generated from precipitation, it is reasonable to use yearly precipitation frequency in the simulated basin as a surrogate of yearly streamflow frequency to judge whether one specific year is a wet year or dry year.

The manuscript has been revised accordingly to enhance how our findings could support the application of this method in data-sparse basins.

*Technical questions:*
*1) I don't agree that "However, none of these observations has the capability of streamflow data for constraining hydrological model parameters." The soil moisture, ground water level can also be used in hydrological calibration and constraining model parameters (e.g., Immerzeel and Droogers. 2008. Calibration of a distributed hydrological model based on satellite evapotranspiration. Journal of Hydrology, 349(3): 411-424.).*
**Response:**

What we want to express here is that the other type of the observations are useful for model calibration, but they are not as useful as streamflow data. The sentence has been revised to avoid confusion.

*2) Through I agree with you that "it is difficult to apply measured values to hydrological models directly", I don't think these parameters are conceptual without explicit physical meaning (e.g., CH_K2, SOL_AWC, SOL_K). They have explicit physical meanings.*
**Response:**
We agree with this comment, the three parameters mentioned by the editor have explicit physical meanings.

*3) Check the language throughout this manuscript. For example, Line 14: period(s); Line 22: year(s)…*
**Response:**
Following this comment, the language has been checked to avoid grammatical errors.

*4) What are the criteria to distinguish a good model performance? How to tell the model performance (e.g., calibrated using one-month/three-month streamflow data) is as good as the one calibrated using 3-year? The meaning of U combining the P_factor with the R_factor should be clarified. For example, the model performance of "one year 2003 or 2004" is not superior significantly than that of "one month" in Table 5.*
**Response:**
After considering this comment carefully, we realize that our idea of combing P_factor and R_factor to make a single index to quantify the simulation uncertainty is not proper, as the new index U has not explicit meaning and is hard to understand and misleading. In the revised manuscript, the simulation uncertainty will be quantified by P_factor and R_factor directly. P_factor stands for the percentage of observations covered by the uncertainty band, which ranges from 0 to 1. R-factor stands for the relative width of the uncertainty bands, which ranges from 0 to infinity. Higher R-factor combined with lower P_factor represents lower simulation uncertainty.

In the revised manuscript, the performance of each calibration is quantified by three indexes computed for the validation period: the Nash-Sutcliffe Efficiency (NSE) coefficient of simulated streamflow corresponding to the best performed parameter set in the calibration period, the P_factor and the R_factor. We put more weight on NSE and P_factor, as they are most explicit and values are easy to be understood. After NSE and P_factor, then R_factor will be considered and less weight are put. Under such philosophy, the concern about the model performance of "one year 2003 or 2004" is not superior significantly than that of "one month" in Table 5 of Heihe Basin in the original manuscript is answered as follows:

As shown in Figure 10 of the revised manuscript, for the validation period the NSE of "one month"(0.69) is lower than "2003"(0.78) and "2004"(0.72). And the P_factor of "one month"(0.27) is also lower than "2003"(0.65) and "2004"(0.51). Therefore, "2003" and "2004" are superior to "one month"

This strategy of evaluation has been used in the new manuscript and revisions have been made accordingly. In the revised manuscripts, instead of tables, figures are employed to show the result of calibrations more concisely.

*6) About the GLUE equation, your approach calculates the posterior distribution at each point first and then gets the 2.5% and 97.5%, while reviewer's approach computes the 2.5% and 97.5% directly. I think both methods are correct.*

**Response:**

Thank you very much for clarifying this issue.

*(7) Location of these two watersheds should be identified in Fig. 1. And Figure 1 and Figure 2 can be merged into Figure 1. The hydrological stations used in this study should be included.*

**Response:**

The locations of basins being studied in the research has been shown in Figure 1 of the revised manuscript. Also in this figure, for each basin, a map showing the hydrological stations being simulated has been included.

*(8) Figure 6: the unit should follow "HESS author instruction".*

**Response:**

The unit in Figure 5 and 6 (Figure 7a and 7c in the new manuscript) has been revised according to HESS author instruction.

**Responses to reviewer #1**

*The paper shows interesting results on distributed hydrological model calibration, in which the authors demonstrate that the SWAT model can be satisfactorily calibrated using 1-6 month daily discharge observation, that is much shorter than normally used for calibration. It can be a large contribution to hydrological modeling for ungauged or poorly gauged basins where long term observation is not available.*

*There are two comments and recommendations:*
*1. The major point of this paper is that a hydrological model can be successfully calibrated even based on a short term observation and wet conditions for both period and basin are preferable for effective calibration. It may be true but I wonder if it happens by chance. The authors discuss meteorological conditions of calibration years (2005-2007 for Jinjiang basin and 2003-2005 for Heihe basin) but do not discuss the conditions of validation years. If the study basins were wet in validation years, it is quite reasonable that short observation for wet period can provide successful calibration, while it is truly surprising if it can provide a good result even for the case that validation years are dry. I would like to recommend the authors to add plots for validation years to cumulative distribution shown in Figs 5 and 6 and discuss more about the conditions of validation years in relation to the conditions of calibration periods.*

**Response:**

Two more basins, one wet and one dry, are added to the study, in order to future prove the effectiveness of our approach. And the results from the two new basins are quite similar to the two old ones, indicating our conclusions are solid. From the finding of the four basins, it is shown that when using one-year data for calibration, for the two wet basins, whether the year is wet or dry have little influences on model calibration, i.e., their information content for model calibration are at similar level as using three-year data. However, in the two arid basins, the influence is significant. Data from wet years have better performances.

The validation years are also added to the cumulative distribution as shown in Figure R1 and R2. For the Heihe Basin (Figure R1), the validation years (2006, 2007 and 2008) are all wet years. Generally, as shown in many studies, model performance in the validation period will decrease, compared with calibration period. However, for the benchmark calibration of the Heihe basin, the performance between validation and calibration period does not show much difference, as shown in Figure 10 of the revised manuscript. This phenomenon may come from the fact revealed by Figure R1 that, the three validation years (2006, 2007, and 2008) are all wet years. And two of the three calibration years (2003, 2005) are also wet years. In this case of dry Heihe Basin, calibration data may contains more information for parameters identification in the data of wet years (2003, 2005) than in the data of dry year (2004).

[Figure]

Figure R1 Cumulative distribution of annual streamflow of Heihe Basin (Yingluoxia station) for the period of 1960 to 2008

For the benchmark calibration in the Jinjiang Basin, compared with calibration period, model performance in the validation period decreases. Figure R2 revealed that the two validation years (2008, 2009) are dry years. The three calibrations years is average (2005) and wet year (2006, 2007). The decrease in model performance is consistent with other studies (e.g. Todorovic and Plavsic 2016), in which model efficiency also decrease when calibration period is wetter than validation period. When using only one year data for calibration, the performances of these three single year in validation period are similar to the three-year data used in benchmark calibration.

[Figure]

Figure R2 Cumulative distribution of available annual streamflow at Jinjiang Basin (Shilong station) for the period of 1958 to 2009

*2. I assume that the wet period of the Heihe basin is from April till September and expected that the calibration based on the six months from Apr. 2004 till Sep. 2004 was capable of giving a good result, but Table 5 shows that no behavioral parameter sets were obtained in this period although many behavioral sets were obtained for the two dry periods from 2003 to 2004 and from 2004 to 2005. This is different from the tendency that is found from other calibrations. It would lead to deeper understanding if the authors could give clear explanations for this exceptional case.*

**Response:**

The Heihe basin is an inland basin in the arid northwest China. Most rainfall occurs in the summer

season. About 75% of total annual streamflow come from the wet period from April to September. As shown in Figure R1, 2004 is an extremely dry year. Compared with normal year, either in wet season or in dry season of 2004, the average streamflow decreased. Considering the big contribution to total annual streamflow, the degree of decreases of streamflow in the rainy season has high possibility to be bigger than the dry season. The runoff generation mechanism in this wet season with extremely low streamflow is very different from normal situation which made the model cannot capture the essence of variation in streamflow, therefore none of the randomly generated 10,000 parameter sets can reproduce the hydrograph of this wet season with acceptable accuracy. This is our explanation about why no parameter set was identified as behavioral ones using data of wet season in 2004 and it has been added to the revised manuscript.

**References:**

Todorovic, A. and Plavsic, J.: The role of conceptual hydrologic model calibration in climate change impact on water resources assessment, J. Water Clim. Change, 7 (1), 16-28, 2016

**Response to reviewer #2:**

*Two watersheds is not enough to conclude your study provides general conclusions! There are groups that use thousands of watersheds, look up large sample hydrology, for example:*
*http://meetingorganizer.copernicus.org/EGU2015/session/18271*
*http://www.hydrol-earth-syst-sci.net/18/463/2014/hess-18-463-2014.html*

**Responses:**

To make conclusions from this study more general, two more basins, a wet one and a dry one are added to this study and used for evaluating the proposed method. The findings from the new basins are quite similar to the two old ones, indicating the conclusions drown by us are generally solid.

*The contribution of the paper is not clear. Such analysis on the quality of calibration data dates back to 1996 (http://www.sciencedirect.com/science/article/pii/0022169495029184) and republication in HESS is not justified!*

**Response:**

The model used in Yapo *et al.*(1996) is a conceptual rainfall-runoff model. In comparison, our study focuses on *physically-based distributed hydrological models*. This is one major difference between our study and Yapo *et al.* (1996). These two types of models face the same problem: lacking of streamflow data for calibrations in ungauged basins. For conceptual rainfall-runoff models, effectiveness of using short period of streamflow data to calibrate the model has been demonstrated in several papers, such as the one Yapo et al.(1996) mentioned by the reviewer, also Perrin et al.(2006), Beven and Tada (2012). However, based on our knowledge, such approach has not been widely tested for physically-based distributed hydrological models. The contributions of this paper are:

Firstly, it is demonstrated that using short period of streamflow data has the possibility to be useful for calibrating physically-based distributed hydrological models, like conceptual model. In the two wet basins, it is possible to only using 1-month data to calibrate the model effectively. In the two dry basins, using 6-month data is possible. These results are different from the common understanding that data of several years are needed for calibrations of such model. The paper could inspire more researchers to think about using such dataset to calibrate distributed hydrological models in basins lacking of streamflow data and test it in more basins. In the past, this approach didn't draw much attention for solving the calibration problem of distributed model in ungauged basins.

Secondly, this study is valuable for evaluating the effectiveness of short period data for model calibration in real world application. Our results show that, the phenomenon that some parameter sets are identified behavioral ones based on the comparison between simulation and observations could be considered as one evidence for making the judgement that the short period data. However, such judgement should be made with careful consideration as our study also shows that it may not be true when the number of the observations is too low or data are observed in dry years or dry period. It may only be valid when using data with a length of several months and observed in rainy season or wet years. To get more general knowledge about when the observations are most informative for model calibration, more researches similar to our studies should be conducted. Based on our findings, the relationship between wetness level of short period data (i.e., the records were observed in a wet year or dry year and in rainy season or dry season) and their effectiveness for parameter calibration are worthy to be explored in this kind of future studies.

*There is no figure provided of how calibration of SWAT with limited data translates into model simulation! How do I know 1 month of data is enough for calibration if I don't see how the model works graphically? NSE is certainly not enough!*

**Response:**

Thank you for this suggestion. For the Jinjiang Basin, a figure showing the simulated hydrograph corresponding to the calibration using data of one month will be added to the revised manuscript to demonstrate the effectiveness of using such short period of data for calibration. The figure is also shown here (Figure R3 below). For the validation period (2008 to 2009), it is shown that the best simulations of ensemble predictions corresponding to the calibrations using three year data (2005 to 2007) and one month data (July 2006) is quite similar visually. As a response to the concern about NES, we computed the Mean Absolute Error (MAE) of best simulations in low flow period (September to next March) and high flow period (April to September). In high flow period, the MAE of the best simulation corresponding to calibration using three-year data and one-month-data is 66.3 $m^3$/s and 63.3$m^3$/s, respectively. In low flow period, the MAE of the two best simulations is 36.4$m^3$/s and 43.1$m^3$/s, respectively. Generally, similar performance level is achieved by the two best simulations. These results could support our conclusion that in the Jinjiang Basin, it is possible that one month data is informative to calibrate the SWAT model effectively.

[Figure]

Figure R3 Observed streamflow of 2008 to 2009(dashed red line), best simulations of ensemble prediction for the validation period (2008-2009) corresponding to calibration based on streamflow data of 2005 to 2007 (solid black line) and July 2006(solid green line) in the Jinjiang Basin.

*Page 2, lines 1-5: I don't agree with your statement that models like SWAT are able to predict droughts and floods! Droughts and floods respond to climatic forcings and climatic models are used to forecast them, certainly not SWAT!*

**Response:**

As mentioned by the reviewer, hydrological models, like SWAT, use climatic data as forcing data to predict streamflow in the river. When climatic forecasting are available, hydrological models can employ it as input and predict hydrological drought and flood, from the perspective of quantity of streamflow.

*Page 2, line 8-9: "Most parameters of hydrological models are conceptual without explicit physical meaning, which makes it necessary to identify parameter values through model calibration based on streamflow data". This refers to conceptual models mostly. Physically based distributed are supposed*

*to have parameters with clear physical meaning, that can ideally be measured in the field.*

**Response:**

We agree with the reviewer's opinion. Although values of parameters with explicit physical meaning can be measured, the scale of measurement and model simulation is different, which makes it difficult to apply measured values to hydrological models directly. Also, such measurements require intensive field survey, which are not available in most researches. Therefore, usually model parameters of physically-based model are also obtained from model calibration based on streamflow data. This explanation has been added to the manuscript.

*"Many recent works have focused on using in situ or remote sensing observations of hydrological processes other than streamflow for model calibration, e.g., soil moisture (e.g., Silvestro et al., 2015; Vrugt 20 et al., 2002), evapotranspiration (Vervoort et al., 2014; Winsemius et al., 2008), groundwater level(e.g., Khu et al.,2008)." My understanding is that since streamflow measurements are not available, one can alternatively use other variables such as soil moisture, ground water table and evapotranspiration as calibration data. This is certainly not the case, since measuring these variables is much more difficult and costly than streamflow. I suggest you phrase your sentences more carefully to avoid such confusions.*

**Response:**

Our intention is not to make new observations of other variables of hydrological cycles, but to make best use of available data of such variables. There are cases that in some basins, streamflow data is unavailable, but observation data of the other variables are available. In such situation, the available data maybe valuable for model calibration. To avoid confusions, these sentences will be revised based on above understanding.

*Page 3, line 4: What do you mean by "changing environment"?*

**Response:**

This term of "changing environment" comes from the paper of Montanari et al. (2013), which introduces the IAHS Scientific Decade 2013–2022 " Panta Rhei—Everything Flows". It means the all factors that influence hydrological system, including both changes in nature (e.g., climate changes) and society (e.g., global population growth).

*Page 3, lines 4-6: You argue "For hydrological simulations or predictions in changing environments, physically-based distributed hydrological models are usually preferred, because of their better description of the spatial heterogeneity and details of the water cycle at the basin scale (Finger et al, 2012; Wu and Liu, 2012)." I understand that some physical modelers would make such arguments, but it is certainly a debated issue, so I wouldn't make such strong claims. This being said, in a changing climate, even physically based models are not proven to be working properly. The argument made by developers of physically based models is that since they use specific description of the watersheds, their models can handle land-use change (change of physical characteristics of watersheds). This also needs a lot of research still.*

**Response:**

We agree with the reviewer's opinion about the comparison between conceptual and physically based hydrological model. The sentences will be revised in the new version of manuscript to express our understanding more precisely.

*Equation 2 is all WRONG! You want to use an objective function of NSE, do, but you can't call it a likelihood function and use it as in the Bayes theorem! There is no scaling in Bayes law! You may call this weight, but not posterior likelihood.*

**Response:**

For GLUE, the term of likelihood is used in a very general sense, as described by developer of GLUE, Keith Beven, in the paper of Beven and Binley (1992): the likelihood function in GLUE works as a fuzzy, belief, or possibilistic measure of how well the model conforms to the observed behavior of the system, and not in the restricted sense of maximum likelihood theory. The likelihood measure quantifies the difference between simulation and observations. The only requirement for a likelihood measure is that it should be assigned as zero for all parameter sets that cannot reproduce the observations and should increase monotonically as the performance rises. Based on the theory of GLUE, in our opinion, using NSE as a likelihood function is proper in our study. Also the Bayes equation is employed in GLUE in a general sense. Equation 2 has been used in many studies related to GLUE, such as Freer and Beven (1996), Beven and Freer (2001).

*I have a hard time with equation 3 also! Weights (or as you call them posterior likelihoods) are calculate based on overall performance of the model (t=1:N), but are used at each time step to estimate the cumulative probability of streaamflow. This is not right! You want to estimate the 95% uncertainty range, take the 2.5 and 97.5th percentiles of your streamflow simulation ensemble.*

**Response:**

For application of GLUE, the posterior likelihood is computed based on the overall model performance in the period when observations are available to compare model behavior with observations. Then at each time step, the posterior likelihood is used to estimate cumulative probability of streamflow. This is the way how GLUE estimates simulation uncertainty, and equation 3 can be found in many studies using GLUE.

*Page 6, lines 4-6: I don't understand this sentence, and it is critical to evaluate the methodology proposed in this paper: "As a first trial for the distributed model, we sought to explore the highest possible performance using certain short lengths of records, not the general performance of specific lengths."*

**Response:**

What we try to express is that, as an initial trial for showing the potential of the method for distributed models, we sought to explore whether there are records of certain short length or number of continuous daily observations could achieve similar performance as benchmark calibration, not to determine whether all records with that specific length can calibrate the model effectively..

This explanation will be added to the revised version of paper.

*The entire section 2.4 is poorly written and methodology is poorly described, making it really difficult to assess the paper.*

**Response**:

This section has been rewritten in the new manuscript to make it easier understood.

*Page 7, line 2: You admit that it is very time consuming to calibrate SWAT using GLUE. Why not using more intelligent calibration approaches like Markov Chain Monte Carlo? It has been shown in the literature that MCMC is orders of magnitude more efficient than GLUE.*

**Response:**

For daily-step model simulation of several years, it is common for lumped conceptual models to finish the simulation within one second. However, for distributed model like SWAT, usually, one such model run may need several minutes. We have no information about prior distribution of model parameters, based on which, parameter sets will be generated randomly. The uniform distribution is used as the prior distribution of each parameter. In such situation, implementing GLUE with Latin hypercube sampling is usually preferred. This strategy has been used in many literatures related to GLUE and SWAT simulation.

*Page 7 line 11: "Kim and Kaluarachchi (2009) and Yapo et al. (1996) showed that data from high-flow periods are more informative than data from low-flow periods for model calibration, because most model modules are activated in high-flow periods." I tend to disagree! It depends on your performance metrics, if you use NSE which is sensitive to peak flows, then yes you are right! But if you use metrics such as baseflow index this is not going to hold. Different processes of a model are activated under different forcings, so you can't simply ignore several processes and focus on the one (few) process(es) that are activated at the wet condition.*

**Response:**

We agree with the reviewer that the statement of "because most model modules are activated in high-flow period" is improper and it will be deleted from the manuscript.

*Page 7 line 24: Uncertainty bound not band! Correct throughout the manuscript.*

**Response:**

Uncertainty bounds are two lines consisted of the 2.5% and 97.5% quantiles of simulated streamflow in each time step. The uncertainty band is the area bounded by these two lines in hydrograph. Both term of "uncertainty bound" and "uncertainty band" (e.g., Beven and Benley, 1992; Yang et al., 2008) can be used when applying GLUE for uncertainty analysis.

*Page 8, lines 2-3: "For the 1-year period, all three calibrations performed similarly to the benchmark calibration, and the dataset for 2006 even outperformed the benchmark" It is interesting and concerning that a shorter calibration period provides a higher performance. It requires explanation as to how it happened! You can't just leave it like that which might spuriously suggest smaller calibration period is sometimes even better! Here are my thoughts: 1. You are not using a consistent period to evaluate your model! 2. Your calibration approach did not converge to the right posterior distribution (as might happen with GLUE) 3. Your data includes some misinformation, meaning not only it doesn't provide any good information to constrain model parameters, but also it misguides the model! In the cases of 1 & 2, extra data can only be redundant and cannot deteriorate the performance of the model!*

**Response:**

The evaluation of each calibration is based on judging model performance in the validation period. The validation period is made to be same for all calibrations in each basin. For the Jinjiang Basin, the validation period is 2008 to 2009. For the Heihe Basin, the validation period is 2006 to 2008.

Streamflow data are obtained from the water administrative department in local government and have used in many studies. The data quality is guaranteed. We apologize that the statement of "the dataset for 2006 even outperformed the benchmark" is misleading. It is only based on the NSE of best performed behavioral parameter set. Another important aspect of the evaluation is simulation uncertainty. It is quantified by the index of "U" as defined in the original manuscript. The U was computed from the P_factor (percentage of observations embraced by the 95% prediction intervals) and R_factor (a measure of the average width of 95% simulation intervals). Our intention of using U was trying to describe simulation uncertainty using one single index. However, after carefully consideration, we feel that the meaning of U value is confusing and decide to using P_factor and R_factor directly to describe simulation uncertainty. As shown in the Figure 6 of the revised manuscript, it is indicated that the simulation uncertainty of calibration using data of 2006 is a little bit higher than benchmark calibration. So we agree that the statement of "the dataset for 2006 even outperformed the benchmark" is improper and it is not our intention to conclude that using shorter period of calibration is better than using long period of observations. The method presented in this study is only expected to be useful for model calibration in data-sparse basins where streamflow data of several years are unavailable. The statement of "the dataset for 2006 even outperformed the benchmark" will be deleted.

*Page 8, lines 14-16: "The calibration using the 1-month dataset still achieved similar performance to benchmark calibration. Thus, it is indicated that in the Jinjiang Basin, it is possible to calibrate the SWAT model effectively using only 1-month's continuous daily observations of streamflow." This claim is rather strange to me! One month is enough to capture all the processes? Some processes might not even be activated in one month! Again, this is because you focused all your attention on NSE, and what is most important in NSE is the high peaks. So if you activate the processes that reproduce the high peaks, you get a good performance. This doesn't mean one month is enough to calibrate a model!*

**Response:**

We apologize that we didn't give enough details for the evaluations of calibration using 1-month data in the manuscript. As our responses to previous comments, from simulated hydrograph and all model performance indexes (NSE, P_factor and R_factor, MAE for both low and high flow period) in the validation period, it is indicated that the calibrated model corresponding to 1-month calibration data performs similar to the benchmark calibration.

We realize that our expression about the results using 1-month data is too strong and confusing. In the revised manuscript, instead of the original expression, we will conclude that it is possible that 1-month's continuous daily observations can contain much of the information content of 3-year continuous streamflow data for model calibration.

All the other revisions suggested by the editor and reviewers are specified in the responses to the comments in the previous section of this pdf file.

[revised manuscript text omitted]
 2007, green line), one month(July 2006, blue line), one week( July 14 to 20, 2006, black line) and in situ observations (red dashed line).**

[Figure]

Figure 9: Model performance for the calibrations using short-period data in Donghe Basin.

[Figure]

**Figure 10: Model performance for the calibrations using short-period data in Heihe Basin.**

[Figure]

[Figure]

**Figure 11: Model performance for the calibrations using short-period data in Yalongjiang Basin**

[Figure]

**Figure 5: Cumulative distribution of available annual streamflow at Shilong station for the period of 1958 to 2009**

[Figure]

**Figure 6: Cumulative distribution of annual streamflow at Yingluoxia station for the period of 1960 to 2008Table 1 Main characteristics of the four basins being studied**

| Basin | Streamflow station | Area (km2) | Climate | Annual Rainfall (mm) | Annual average temperature(℃) | Ranges of Elevation(m) |
|---|---|---|---|---|---|---|
| Jinjiang | Shilong | 5,629 | Subtropical marine monsoon climate | 1651 | 20 | 50 to 1366 |

| Donghe | Wenquan | 1,089 | Subtropical monsoon climate | 1247 | 18 | 192 to 2569 |
| Heihe | Yingluoxia | 8,843 | Continental monsoon climate | 423 | 6 | 1711 to 4749 |
| Yalongjiang | Ganzi | 3,2535 | Continental plateau climate | 570 | 8 | 3400 to 6021 |

**Table 2 The calibration and validation period for the benchmark calibrations of the four basins**

| Basin | Calibration period | Validation period |
| --- | --- | --- |
| Jinjiang | 2005 to 2007 | 2008 to 2009 |
| Donghe | 2002 to 2004 | 2005 to 2006 |
| Heihe | 2003 to 2005 | 2006 to 2008 |
| Yalongjiang | 2005 to 2007 | 2008 to 2010 |

**Table 13 Short periods for which corresponding data were used for the calibrations for at the stage one of the evaluation**

| Length of the period | Jinjiang Basin | Donghe Basin | Heihe Basin | Yalongjiang Basin |
| --- | --- | --- | --- | --- |
| | 2005 | 2002 | 2003 | 2005 |
| One year | 2006 | 2003 | 2004 | 2006 |
| | 2007 | 2004 | 2005 | 2007 |
| | April 2005 to September 2005 | April 2002 to September 2002 | April 2003 to September 2003 | April 2005 to September 2005 |
| | October 2005 to March 2006 | October 2002 to March 2003 | October 2003 to March 2004 | October 2005 to March 2006 |
| Six months | April 2006 to September 2006 | April 2003 to September 2003 | April 2004 to September 2004 | April 2006 to September 2006 |
| | October 2006 to March 2007 | October 2003 to March 2004 | October 2004 to March 2005 | October 2006 to March 2007 |
| | April 2007 to September 2007 | April 2004 to September 2004 | April 2005 to September 2005 | April 2007 to September 2007 |

10 **Table 2 4 SWAT model parameters being calibrated**


| Name | Description | Initial range |
|---|---|---|
| CN2 | SCS runoff curve number | 20–90 |
| EPCO | Plant uptake compensation factor | 0.01–1 |
| GW_DELAY | Groundwater delay time (days) | 30–450 |
| SLSUBBSN | Average slope length (m) | 10–150 |
| ESCO | Soil evaporation compensation coefficient | 0.8–1 |
| ALPHA_BF | Baseflow recession coefficient | 0–1 |
| OV_N | Manning coefficient for overland flow | 0–0.8 |
| CH_K2 | Hydraulic conductivity in main channel (mm/hr) | 5–130 |
| SOL_AWC | Available soil water capacity (mm $H_2O$/mm Soil) | 0–1 |
| SOL_K | Soil Saturated hydraulic conductivity (mm/hr) | 0–2000 |

**Table  5 Model performance for the benchmark calibration in the two basins**

| | Number of behavioral parameter sets | NSE | | P_factor | | R_factor | |
|---|---|---|---|---|---|---|---|
| | | Calibration | Validation | Calibration | Validation | Calibration | Validation |
| Jinjiang Basin | 2814 | 0.85 | 0.52 | 0.66 | 0.81 | 0.56 | 1.12 |
|  |  |  |  |  |  | | |
| Jinjiang Basin | 2814 | 0.70 | 0.75 | 0.82  | 0.81 | 0.42 | 0.40 |
| Heihe Basin | 1445 | 0.78 | 0.78 | 0.56 | 0.54 | 1.00 | 0.91 |
| Yalongjiang Basin | 1831 | 0.59 | 0.73 | 0.72 | 0.79 | 0.92 | 0.83 |

|  |  |  |  | |  | |
|---|---|---|---|---|---|---|
| | | |  |  |  |  |
|  |  |  |  |  |  |  |
| |  |  |  |  |  |  |

| Length of records | Calibration period | Number of behavioral sets | NSE Calibration | NSE Validation | U Calibration | U Validation |
|---|---|---|---|---|---|---|
| | 2007 | 3888 | 0.88 | 0.58 | -0.15 | 0.32 |
| Six months | Apr. 2005 to Sep. 2005 | 1916 | 0.85 | 0.52 | 0.22 | 0.25 |
| | Oct. 2005 to Mar. 2006 | 492 | 0.88 | 0.62 | 0.07 | 0.13 |
| | Apr. 2006 to Sep. 2006 | 2359 | 0.83 | 0.67 | -0.20 | 0.28 |
| | Oct. 2006 to Mar. 2007 | - | - | - | - | - |
| | Apr. 2007 to Sep. 2007 | 3163 | 0.86 | 0.51 | 0.06 | 0.31 |
| Three months | Jun. 2006 to Aug. 2006 | 2338 | 0.84 | 0.65 | 0.12 | 0.33 |
| One month | Jul. 2006 | 3064 | 0.86 | 0.57 | -0.40 | 0.34 |
| One week | Jul 14 to 20, 2006 | 1370 | 0.87 | 0.40 | -0.11 | 0.38 |

Table 5 Model performance for the calibrations using short-period data in Heihe basin

| Length of records | Calibration period | Number of behavioral sets | NSE | | U | |
|---|---|---|---|---|---|---|
| | | | Calibration | Validation | Calibration | Validation |
| | 2003 | 3311 | 0.88 | 0.78 | 0.41 | 0.51 |
| One year | 2004 | 39 | 0.63 | 0.72 | 0.24 | 0.12 |
| | 2005 | 1282 | 0.79 | 0.78 | 0.47 | 0.41 |
| Six months | Apr. 2003 to Sep. 2003 | 2113 | 0.82 | 0.78 | 0.41 | 0.48 |
| | Oct. 2003 to Mar. 2004 | 843 | 0.91 | 0.54 | 0.32 | 0.50 |
| | Apr. 2004 to Sep. 2004 | - | - | - | - | - |

| | | | | | |
|---|---|---|---|---|---|
| | Oct. 2004 to Mar. 2005 | 84 | 0.81 | 0.44 | 0.36 | 0.46 |
| | Apr. 2005 to Sep. 2005 | 202 | 0.64 | 0.71 | 0.38 | 0.13 |
| Three months | Jun. 2003 to Aug. 2003 | 1195 | 0.75 | 0.72 | 0.43 | 0.45 |
| One month | Aug. 2003 | 46 | 0.63 | 0.69 | 0.34 | 0.48 |
| One week | Aug 8 to 14, 2003 | 1 | 0.52 | 0.71 | - | - |

---

## Editor Decision (ED1)

Dear Authors,

Thank you for your contribution to HESS. However, there are still several questions need to be clarified.

(1) As is suggested by reviewer #2, two watersheds, typical though, are far from enough. If this manuscript is designed to only provide an idea or assumption on "short period calibration of hydrological models", it does not contribute a lot to the hydrological community as many previous researches have already done this using different models though (same comments as reviewer #2). Therefore, I think you may conduct few more case studies, test it and make it convincing. Or if this manuscript focuses on short period calibration of "Physically-based distributed hydrological model", a discussion on the calibration of both conceptual models and physical models is expected. And also their differences in calibration should be discussed. In addition, what can we learn from this idea and how can we use this idea in the calibration of hydrological models for data-scarce basins?

(2) In your responses to reviewer #1, as is concluded that the good model performance for the validation period "may come from the fact that, the three validation years (2006, 2007, and 2008) are all wet years. And two of the three calibration years (2003, 2005) are also wet years. Calibration data may contain sufficient information for parameters identification in wet years." If in this case, the model is useful only when the calibration period and validation period both are wet years or both are dry years. What can we do when only one year's data is available for a data-scarce watershed? We cannot tell if this year is as the same condition as the following/previous years. Therefore, the objective of this study "how the use of limited continuous daily streamflow data might support the application of a physically-based distributed model in data-sparse basins" is not well explained.

**Technical questions:**
1) I don't agree that "However, none of these observations has the capability of streamflow data for constraining hydrological model parameters." The soil moisture, ground water level can also be used in hydrological calibration and constraining model parameters (e.g., Immerzeel and Droogers. 2008. Calibration of a distributed hydrological model based on satellite evapotranspiration. Journal of Hydrology, 349(3): 411-424.).
2) Through I agree with you that "it is difficult to apply measured values to hydrological models directly", I don't think these parameters are conceptual without explicit physical meaning (e.g., CH_K2, SOL_AWC, SOL_K). They have explicit physical meanings.
3) Check the language throughout this manuscript. For example, Line 14: period(s); Line 22: year(s)…
4) What are the criteria to distinguish a good model performance? How to tell the model performance (e.g., calibrated using one-month/three-month streamflow data) is as good as the one calibrated using 3-year? The meaning of U combining the P_factor with the R_factor should be clarified. For example, the model performance of "one year 2003 or 2004" is not superior significantly than that of "one month" in Table 5.
6) About the GLUE equation, your approach calculates the posterior distribution at each point first and then gets the 2.5% and 97.5%, while reviewer's approach computes the 2.5%

and 97.5% directly. I think both methods are correct.

(7) Location of these two watersheds should be identified in Fig. 1. And Figure 1 and Figure 2 can be merged into Figure 1. The hydrological stations used in this study should be included.

(8) Figure 6: the unit should follow "HESS author instruction".